# PI-FLOW: POLICY-BASED FEW-STEP GENERATION VIA IMITATION DISTILLATION

**Hansheng Chen**[1]  **Kai Zhang**[2]  **Hao Tan**[2]  **Leonidas Guibas**[1]  **Gordon Wetzstein**[1]  **Sai Bi**[2]

[1]Stanford University  [2]Adobe Research

https://github.com/Lakonik/piFlow

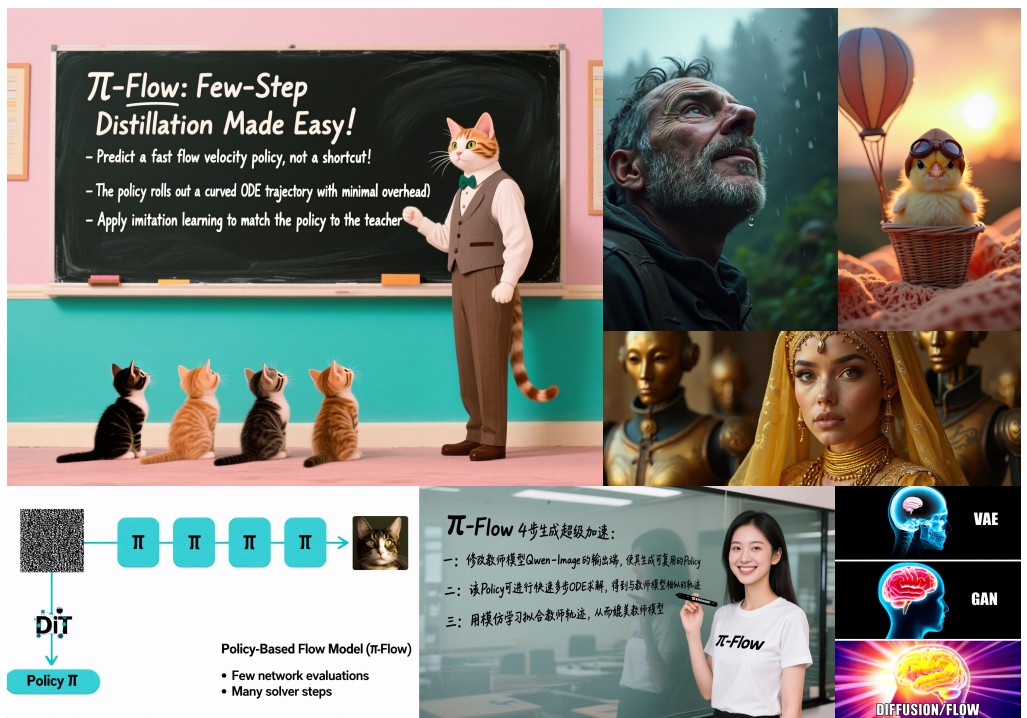

Figure 1: High quality 4-NFE text-to-image generations by $\pi$-Flow, distilled from FLUX.1-12B (top-right three images) and Qwen-Image-20B (all remaining images). $\pi$-Flow preserves the teacher's coherent structures, fine details (e.g., skin and hair), and accurate text rendering, while avoiding diversity collapse (see Fig. 4 for sample diversity).

## ABSTRACT

Few-step diffusion or flow-based generative models typically distill a velocity-predicting teacher into a student that predicts a shortcut towards denoised data. This format mismatch has led to complex distillation procedures that often suffer from a quality–diversity trade-off. To address this, we propose *policy-based flow models* ($\pi$-Flow). $\pi$-Flow modifies the output layer of a student flow model to predict a network-free policy at one timestep. The policy then produces dynamic flow velocities at future substeps with negligible overhead, enabling fast and accurate ODE integration on these substeps without extra network evaluations. To match the policy's ODE trajectory to the teacher's, we introduce a novel imitation distillation approach, which matches the policy's velocity to the teacher's along the policy's trajectory using a standard $\ell_2$ flow matching loss. By simply mimicking the teacher's behavior, $\pi$-Flow enables stable and scalable training and avoids the quality–diversity trade-off. On ImageNet $256^2$, it attains a 1-NFE FID of 2.85, outperforming previous 1-NFE models of the same DiT architecture. On FLUX.1-12B and Qwen-Image-20B at 4 NFEs, $\pi$-Flow achieves substantially better diversity than state-of-the-art DMD models, while maintaining teacher-level quality.

# 1 INTRODUCTION

Diffusion and flow matching models (Sohl-Dickstein et al., 2015; Ho et al., 2020; Song & Ermon, 2019; Lipman et al., 2023; Albergo & Vanden-Eijnden, 2023) have become the dominant method for visual generation, delivering compelling image quality and diversity. However, these models rely on a costly denoising process for inference, which integrates a probability flow ODE (Song et al., 2021) over multiple timesteps, each step requiring a neural network evaluation. Commonly, the inference cost of diffusion models is quantified by the number of function (network) evaluations (NFEs).

To reduce the inference cost, diffusion distillation methods compress a pre-trained multi-step model (the teacher) into a student that requires only one or a few network evaluation steps. Existing distillation approaches avoid ODE integration by taking one or a few shortcut steps that map noise to data, where each shortcut path is predicted by the student network, referred to as a *shortcut-predicting* model. Learning these shortcuts is a significant challenge because they cannot be directly inferred from the teacher model. This necessitates the use of complex training methods, such as progressive distillation (Salimans & Ho, 2022; Liu et al., 2023; 2024), consistency distillation (Song et al., 2023), and distribution matching (Sauer et al., 2024a; Yin et al., 2024b;a; Salimans et al., 2024). In turn, the sophisticated training often lead to degraded image quality from error accumulation or compromised diversity due to mode collapse.

To sidestep the difficulties in shortcut-predicting distillation, we propose a novel *policy-based flow model* ($\pi$-Flow or pi-Flow) paradigm: given noisy data at one timestep, the student network predicts a network-free policy, which maps new noisy states to their corresponding flow velocities with negligible overhead, allowing fast and accurate ODE integration using multiple substeps of policy velocities instead of network evaluations.

To train the student network, we introduce *policy-based imitation distillation* ($\pi$-ID), a DAgger-style (Ross et al., 2011) on-policy imitation learning (IL) method. $\pi$-ID trains the policy on its own trajectory: at visited states, we query the teacher velocity and match the policy's output to it, using the teacher's corrective signal to teach the policy to recover from its own mistakes and reduce error accumulation. Specifically, the matching employs a standard $\ell_2$ loss aligned with the teacher's flow matching objective, thus naturally preserving its quality and diversity.

We validate our paradigm with two types of policies: a simple dynamic-$\hat{\boldsymbol{x}}_0^{(t)}$ (DX) policy and an advanced GMFlow policy based on Chen et al. (2025). Experiments show that GMFlow policy outperforms DX policy and delivers strong ImageNet $256^2$ FIDs at 1- and 2-NFE generation. To demonstrate its scalability, we distill FLUX.1-12B (Black Forest Labs, 2024b) and Qwen-Image-20B (Wu et al., 2025) text-to-image models into 4-NFE $\pi$-Flow students, which achieve state-of-the-art diversity, while maintaining teacher-level quality.

We summarize the contributions of this work as follows:

- We propose $\pi$-Flow, a new paradigm that decouples ODE integration substeps from network evaluation steps, enabling both fast generation and straightforward distillation.
- We introduce $\pi$-ID, a novel on-policy IL method for few-step $\pi$-Flow distillation, which reduces the training objective to a simple $\ell_2$ flow matching loss.
- We demonstrate strong performance and scalability of $\pi$-Flow, particularly, its superior diversity and teacher alignment compared to other state-of-the-art 4-NFE text-to-image models.

# 2 PRELIMINARIES

In this section, we briefly introduce flow matching models (Lipman et al., 2023; Liu et al., 2023) and the notations used in this paper.

Let $p(\boldsymbol{x}_0)$ denote the (latent) data probability density, where $\boldsymbol{x}_0 \in \mathbb{R}^D$ is a data point. A standard flow model defines an interpolation between a data sample and a random Gaussian noise $\boldsymbol{\epsilon} \sim \mathcal{N}(\boldsymbol{0}, \boldsymbol{I})$, yielding the diffused noisy data $\boldsymbol{x}_t = \alpha_t \boldsymbol{x}_0 + \sigma_t \boldsymbol{\epsilon}$, where $t \in (0, 1]$ denotes the diffusion time, and $\alpha_t = 1 - t, \sigma_t = t$ are the linear flow noise schedule. The optimal transport map across all marginal densities $p(\boldsymbol{x}_t) = \int_{\mathbb{R}^D} \mathcal{N}(\boldsymbol{x}_t; \alpha_t \boldsymbol{x}_0, \sigma_t^2 \boldsymbol{I}) p(\boldsymbol{x}_0) \, \mathrm{d}\boldsymbol{x}_0$ can be described by

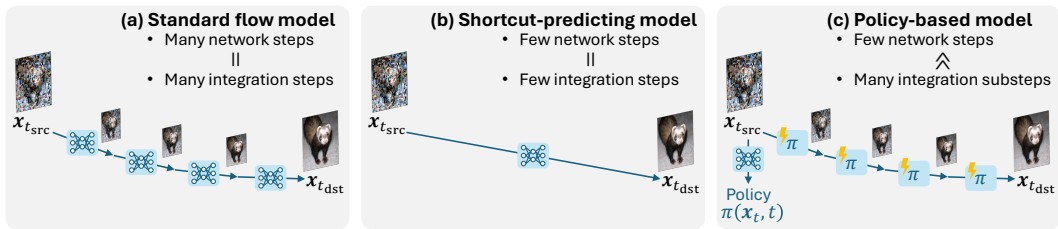

Figure 2: Comparison between (a) standard flow model (teacher), (b) shortcut-predicting model, and (c) our policy-based model. The shortcut-predicting model skips all intermediate states, whereas the our-based model retains all intermediate substeps with minimal overhead.

the following probability flow ODE (Song et al., 2021; Liu, 2022):

$$\frac{\mathrm{d}\boldsymbol{x}_t}{\mathrm{d}t} = \dot{\boldsymbol{x}}_t = \frac{\boldsymbol{x}_t - \mathbb{E}_{\boldsymbol{x}_0 \sim p(\boldsymbol{x}_0|\boldsymbol{x}_t)}[\boldsymbol{x}_0]}{t} = \frac{\boldsymbol{x}_t - \int_{\mathbb{R}^D} \boldsymbol{x}_0 p(\boldsymbol{x}_0|\boldsymbol{x}_t) \, \mathrm{d}\boldsymbol{x}_0}{t}, \tag{1}$$

with the denoising posterior $p(\boldsymbol{x}_0|\boldsymbol{x}_t) := \frac{\mathcal{N}(\boldsymbol{x}_t; \alpha_t \boldsymbol{x}_0, \sigma_t^2 \boldsymbol{I}) p(\boldsymbol{x}_0)}{p(\boldsymbol{x}_t)}$. At test time, the model can generate samples by first initializing the noise $\boldsymbol{x}_1 \leftarrow \boldsymbol{\epsilon}$ and then solving the ODE to obtain $\lim_{t \to 0} \boldsymbol{x}_t$.

In practice, flow matching models approximate the ODE velocity $\frac{\mathrm{d}\boldsymbol{x}_t}{\mathrm{d}t}$ using a neural network $G_{\boldsymbol{\theta}}(\boldsymbol{x}_t, t)$ with learnable parameters $\boldsymbol{\theta}$, trained using the $\ell_2$ flow matching loss:

$$\mathcal{L}_{\boldsymbol{\theta}} = \mathbb{E}_{t, \boldsymbol{x}_0, \boldsymbol{x}_t}\left[\frac{1}{2}\|\boldsymbol{u} - G_{\boldsymbol{\theta}}(\boldsymbol{x}_t, t)\|^2\right], \quad \text{with sample velocity } \boldsymbol{u} := \frac{\boldsymbol{x}_t - \boldsymbol{x}_0}{t}. \tag{2}$$

Since each velocity query requires evaluating the network (Fig. 2 (a)), flow matching models couple sampling efficiency with solver precision. Despite the progress in advanced solvers (Karras et al., 2022; Zhang & Chen, 2023; Lu et al., 2022; 2023; Zhao et al., 2023), high-quality sampling typically requires over 10 steps due to inherent ODE truncation error, making it computationally expensive.

## 3  $\pi$-FLOW: POLICY-BASED FEW-STEP GENERATION

In $\pi$-Flow, we define the policy as a *network-free* function $\pi \colon \mathbb{R}^D \times \mathbb{R} \to \mathbb{R}^D$ that maps a state $(\boldsymbol{x}_t, t)$ to a flow velocity. A policy can be network-free if it only needs to describe a single ODE trajectory, which is fully determined by its initial state $(\boldsymbol{x}_{t_{\mathrm{src}}}, t_{\mathrm{src}})$ with $t_{\mathrm{src}} \geq t$. In this case, the policy for each trajectory must be dynamically predicted by a neural network conditioned on that initial state $(\boldsymbol{x}_{t_{\mathrm{src}}}, t_{\mathrm{src}})$. We therefore adapt a flow model to output not a single velocity, but an entire dynamic policy that governs the full trajectory. Formally, define the policy function space $\mathcal{F} := \left\{ \pi \colon \mathbb{R}^D \times \mathbb{R} \to \mathbb{R}^D \right\}$. Then, our goal is to distill a policy generator network $G_{\boldsymbol{\phi}} \colon \mathbb{R}^D \times \mathbb{R} \to \mathcal{F}$ with learnable parameters $\boldsymbol{\phi}$, such that $\pi(\boldsymbol{x}_t, t) = G_{\boldsymbol{\phi}}(\boldsymbol{x}_{t_{\mathrm{src}}}, t_{\mathrm{src}})(\boldsymbol{x}_t, t)$.

As shown in Fig. 2 (c), $\pi$-Flow performs ODE-based denoising from $t_{\mathrm{src}}$ to $t_{\mathrm{dst}}$ via two stages:

- A *policy generation step*, which feeds the initial state $(\boldsymbol{x}_{t_{\mathrm{src}}}, t_{\mathrm{src}})$ to the student network $G_{\boldsymbol{\phi}}$ to produce the policy $\pi$, i.e., $\pi \leftarrow G_{\boldsymbol{\phi}}(\boldsymbol{x}_{t_{\mathrm{src}}}, t_{\mathrm{src}})$.
- Multiple *policy integration substeps*, which integrates the ODE by querying the policy velocity over multiple substeps, obtaining a less noisy state by $\boldsymbol{x}_{t_{\mathrm{dst}}} \leftarrow \boldsymbol{x}_{t_{\mathrm{src}}} + \int_{t_{\mathrm{src}}}^{t_{\mathrm{dst}}} \pi(\boldsymbol{x}_t, t) \, \mathrm{d}t$.

Unlike previous few-step distillation methods, $\pi$-Flow decouples network evaluation steps from ODE integration substeps. This allows it to combine the key advantages of two paradigms: it performs only a few network evaluations for efficient generation, similar to a shortcut-predicting model, while also executing dense integration substeps, just like a standard flow matching teacher. Thanks to its teacher-like ODE integration process, a $\pi$-Flow student offers unprecedented advantage in training, as we can now follow well-established imitation learning (IL) approaches to directly match the policy velocity $\pi(\boldsymbol{x}_t, t)$ to the teacher velocity $G_{\boldsymbol{\theta}}(\boldsymbol{x}_t, t)$, as discussed later in § 4.

To identify the appropriate function classes of student policies for fast image generation, we need to consider the following requirements:

- **Efficiency.** The policy should provide closed-form velocities with minimal overhead, so that rolling out dense (e.g., 100+) substeps incurs negligible cost compared to a network evaluation.

- **Compatibility.** The policy should have a compact set of parameters that can be easily predicted by the student $G_\phi$ with standard backbones (e.g., DiT (Peebles & Xie, 2023)).
- **Expressiveness.** The policy should be able to approximate a complicated ODE trajectory starting from a certain initial state $x_{t_{\mathrm{src}}}$.
- **Robustness.** The policy should be able to handle trajectory variations that arise from perturbations to the initial state $x_{t_{\mathrm{src}}}$. For instance, a suboptimal student network will produce an erroneous mapping from $x_{t_{\mathrm{src}}}$ to $\pi$. This introduces the randomness that the policy needs to accommodate throughout the rollout. Consequently, the policy function should adapt its velocity output to variations in its state input $x_t$, which is a challenging requirement for network-free functions.

## 3.1 DYNAMIC-$\hat{x}_0^{(t)}$ POLICY

We introduce a simple baseline policy called dynamic-$\hat{x}_0^{(t)}$ policy (*DX* policy). DX policy defines $\pi(x_t, t) := \frac{x_t - \hat{x}_0^{(t)}}{t}$, where $\hat{x}_0^{(t)}$ approximates the posterior moment $\mathbb{E}_{x_0 \sim p(x_0 | x_t)}[x_0]$ in Eq. (1). Along a fixed trajectory starting from an initial state $(x_{t_{\mathrm{src}}}, t_{\mathrm{src}})$, the posterior moment is only dependent on $t$. Therefore, we first predict a grid of $\hat{x}_0^{(t_i)}$ at $N$ evenly spaced times $t_1, ..., t_N \in [t_{\mathrm{dst}}, t_{\mathrm{src}}]$ by a single evaluation of the student network $G_\phi(x_{t_{\mathrm{src}}}, t_{\mathrm{src}})$. This is achieved by expanding the output channels of the student network and performing $u$-to-$x_0$ reparameterization. Then, for arbitrary $t \in [t_{\mathrm{dst}}, t_{\mathrm{src}}]$, we obtain the approximated moment $\hat{x}_0^{(t)}$ by a linear interpolation over the grid.

Apparently, DX policy is fast, compatible, and expressive enough so that any $N$-step teacher trajectory can be matched with $N$ grid points. However, its robustness is limited because $\hat{x}_0^{(t)}$ is not adaptive to perturbations in $x_t$.

## 3.2 GMFLOW POLICY

For stronger robustness, we incorporate an advanced GMFlow policy based on the closed-form GM velocity field in Chen et al. (2025). GMFlow policy expands the network output channels to predict a factorized Gaussian mixture (GM) velocity distribution $q(u | x_{t_{\mathrm{src}}}) = \prod_{i=1}^{L} \sum_{k=1}^{K} A_{ik} \mathcal{N}(u_i; \mu_{ik}, s^2 I)$, where $A_{ik} \in \mathbb{R}_+$, $\mu_{ik} \in \mathbb{R}^C$, $s \in \mathbb{R}_+$ are GM parameters predicted by the network, $L \times C$ factorizes the data dimension $D$ into sequence length $L$ and channel size $C$, and $K$ is a hyperparameter specifying the number of mixture components. Intuitively, the student network $G_\phi$ maps the initial state $x_{t_{\mathrm{src}}}$ to multiple denoising modes that parameterize the GMFlow policy. The policy then enables a closed-form velocity expression at future state $(x_t, t)$ for any $0 < t < t_{\mathrm{src}}$ (see § F for details). The speed and compatibility of GMFlow has already been discussed in Chen et al. (2025), thus we focus on analyzing its expresiveness and robustness.

**Expressiveness.** With the $L \times C$ factorization, each individual $C$-dimensional GM needs to be expressive enough to approximate a $C$-dimensional chunk of the teacher trajectory. In § E, we rigorously prove the following theorem, demonstrating GMFlow's expressiveness.

**Theorem 1** (A GMFlow policy with $K = N \cdot C$ can accurately approximate any $N$-step trajectory)**.** Given pairwise distinct times $t_1, \ldots, t_N \in (0, 1]$ and vectors $x_{t_n}, \dot{x}_{t_n} \in \mathbb{R}^C$ for $n = 1, \ldots, N$, there exists a GM parameterization of $p(x_0)$ with $N \cdot C$ components, such that $\dot{x}_{t_n}$ can be approximated arbitrarily well using Eq (1) at $t = t_n$ for every $n = 1, \ldots, N$.

In practice, we can use $K \ll N \cdot C$ (e.g., $K = 8$) since the teacher trajectory is mostly smooth. More analysis of GMFlow hyperparameters are presented in § C.1.

**Robustness.** GMFlow is highly robust against trajectory perturbation due to its probabilistic origin. Unlike DX policy, GMFlow models a fully dynamic denoising posterior (Eq. (23)) dependent on both $x_t$ and $t$. Leveraging its robustness, the policy can be flexibly altered via GM dropout in training (§ 4) and GM temperature in inference (§ B.1), both improving generalization performance.

---

**Algorithm 1:** On-policy $\pi$-ID.

**Input:** $NFE$, teacher $G_{\boldsymbol\theta}$, student $G_{\boldsymbol\phi}$, condition $\boldsymbol c$

1  Sample $t_{\mathrm{src}}$ from $\left\{ \frac{1}{NFE}, \frac{2}{NFE}, \cdots, 1 \right\}$
2  Initialize $\boldsymbol x_{t_{\mathrm{src}}}$ (data-free or data-dependent)
3  $\pi \leftarrow G_{\boldsymbol\phi}(\boldsymbol x_{t_{\mathrm{src}}}, t_{\mathrm{src}}, \boldsymbol c)$
4  $\pi_{\mathrm D} \leftarrow \mathrm{stopgrad}(\pi)$
5  $\mathcal{L}_{\boldsymbol\phi} \leftarrow 0$
6  **for** *finite samples* $t \sim U\!\left(t_{\mathrm{src}} - \frac{1}{NFE}, t_{\mathrm{src}}\right)$ **do**
7  $\quad \boldsymbol x_t \leftarrow \boldsymbol x_{t_{\mathrm{src}}} + \int_{t_{\mathrm{src}}}^{t} \pi_{\mathrm D}(\boldsymbol x_t, t)\,\mathrm{d}t$
8  $\quad \mathcal{L}_{\boldsymbol\phi} \leftarrow \mathcal{L}_{\boldsymbol\phi} + \frac{1}{2}\|G_{\boldsymbol\theta}(\boldsymbol x_t, t, \boldsymbol c) - \pi(\boldsymbol x_t, t)\|^2$
9  $\boldsymbol\phi \leftarrow \mathrm{Adam}(\boldsymbol\phi, \nabla_{\boldsymbol\phi}\mathcal{L}_{\boldsymbol\phi})$  *// optimizer step*

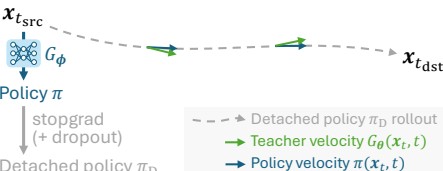

Figure 3: On-policy flow imitation distillation. Intermediate states are sampled along the detached policy rollout, where the loss matches the policy to the teacher.

# 4  $\pi$-ID: POLICY-BASED IMITATION DISTILLATION

With the policy rollout sharing the same format as the teacher's ODE integration, it is straightforward to adopt imitation learning to learn the policy by directly matching the policy's velocity to the teacher's velocity. In this section, we introduce a simple policy-based imitation distillation ($\pi$-ID) algorithm based on DAgger-style (Ross et al., 2011) on-policy imitation.

On-policy imitation learning is robust to error accumulation since it trains the policy on its own trajectory, allowing the teacher's corrective signal to steer a deviating trajectory back on track. As shown in Fig. 3 and Algorithm 1, for a time interval from $t_{\mathrm{src}}$ to $t_{\mathrm{dst}}$ (i.e., a 1-NFE segment), we first feed the initial state $(\boldsymbol x_{t_{\mathrm{src}}}, t_{\mathrm{src}})$ to the student network $G_{\boldsymbol\phi}$ to obtain the policy $\pi$. We then sample an intermediate time $t \in (t_{\mathrm{dst}}, t_{\mathrm{src}}]$ and roll out a *detached* policy $\pi_{\mathrm D}$ from $t_{\mathrm{src}}$ to $t$ using high-accuracy ODE integration (with a small step size of $1/128$), yielding an intermediate state $\boldsymbol x_t$ on the policy trajectory. This state is fed to both the learner policy $\pi$ and the frozen teacher $G_{\boldsymbol\theta}$, which produce their respective velocities. Finally, we compute a standard $\ell_2$ flow matching loss between the two velocities, and backpropagate its gradients through the policy $\pi$ to the student network $G_{\boldsymbol\phi}$. Because the student forward/backward pass dominates compute while policy and teacher queries are relatively cheap, we may repeat the rollout-and-matching step multiple times for additional teacher supervisions. In practice, we sample two intermediate states per student forward pass.

**Data-dependent and data-free $\pi$-ID.** The initial state $\boldsymbol x_{t_{\mathrm{src}}}$ can be obtained via forward diffusion from real data $\boldsymbol x_0$ (data-dependent Algorithm 2), or via $\pi$-Flow's reverse denoising from random noise $\boldsymbol x_1$ (data-free Algorithm 3). Both methods have roughly the same computational cost, and comparable performance, as demonstrated in the experiments (§ 5).

**Error bounds and convergence.** As discussed by Ross et al. (2011), on-policy imitation learning guarantees that the performance of the learned policy is bounded by the teacher's performance plus an error term that scales as $O(n\varepsilon)$, where $n$ is the number of substeps and $\varepsilon$ is the average imitation error per step, which is strictly better than the $O(n^2\varepsilon)$ compounding-error behavior of off-policy behavior cloning. Moreover, the sequence of on-policy iterates converges in performance to the best policy in the function class, under the student's capacity constraint.

# 5  EXPERIMENTS

To demonstrate the versatility of $\pi$-Flow, we evaluate it with three distinct image generation models of different scales and architectures: DiT(SiT)-XL/2 (675M) (Peebles & Xie, 2023; Ma et al., 2024; Vaswani et al., 2017) for ImageNet $256^2$ (Deng et al., 2009) class-conditioned generation, FLUX.1-12B (Black Forest Labs, 2024b) and Qwen-Image-20B (Wu et al., 2025) for text-to-image generation.

## 5.1  IMPLEMENTATION DETAILS

In this subsection, we discuss key implementation details essential to model performance. More training details and hyperparameter choices are presented in § C.

Table 1: 1-NFE generation results of $\pi$-Flow with DX and GMFlow policies on ImageNet. Tested after 40K training iterations. FM stands for standard flow matching.

| Policy | Teacher | FID↓ | IS↑ | Precision↑ | Recall↑ |
|---|---|---|---|---|---|
| DX ($N = 10$) | REPA | 4.73 | 327.6 | 0.781 | 0.514 |
| DX ($N = 20$) | REPA | 4.44 | 329.8 | 0.786 | 0.531 |
| DX ($N = 40$) | REPA | 4.90 | 321.8 | 0.778 | 0.537 |
| GM ($K = 8$) | REPA | **3.07** | 336.9 | 0.789 | **0.572** |
| GM ($K = 32$) | REPA | 3.08 | **341.7** | **0.791** | 0.562 |
| GM ($K = 32$) | FM | **3.65** | **282.0** | 0.797 | **0.533** |
| GM ($K = 32$) w/o dropout | FM | 4.14 | 279.6 | **0.799** | 0.525 |

Table 2: Comparison with previous few-step DiTs on ImageNet.

| Model | NFE | FID↓ |
|---|---|---|
| iCT | 2 | 20.30 |
| iMM | 1×2 | 7.77 |
| MeanFlow | 2 | 2.20 |
| FACM (REPA) | 2 | **1.52** |
| $\pi$-Flow (GM-REPA) | 2 | 1.97 |
| iCT | 1 | 34.24 |
| Shortcut | 1 | 10.60 |
| MeanFlow | 1 | 3.43 |
| $\pi$-**Flow (GM-FM)** | 1 | 3.34 |
| $\pi$-**Flow (GM-REPA)** | 1 | **2.85** |

**GM dropout.** Dropout is a widely adopted technique in supervised/imitation learning and reinforcement learning to improve generalization (Srivastava et al., 2014; Cobbe et al., 2019). For the GMFlow policy, we introduce GM dropout in training to stochastically perturb and diversify $\pi$-ID rollouts to make the policy more robust to potential trajectory variations. Given the GM mixture weights $A_{ik}$ of the detached policy $\pi_D$, we sample a binary mask for each component $k = 1, \cdots, K$ and multiply it into $A_{ik}$ synchronously across all $i = 1, \cdots, L$. The masked weights are then renormalized and used for the detached rollout. By exploring alternative GM modes, this simple technique improves the policy's robustness, yielding better FID on ImageNet $256^2$ (§ 5.2).

**Handling FLUX.1 dev teacher.** On-policy imitation learning assumes the teacher is robust to out-of-distribution (OOD) intermediate states and can steer trajectories back on track. This generally holds for standard flow models with classifier-free guidance (CFG) (Ho & Salimans, 2021), which exhibit error-correction behavior (Chidambaram et al., 2024). However, FLUX.1 dev (Black Forest Labs, 2024b) is a guidance-distilled model without true CFG and is less robust to OOD inputs. To mitigate OOD exposure, we adopt a scheduled trajectory mixing strategy, which rolls out the trajectory using a mixture of teacher and student with a linearly decaying teacher ratio (see § B.2 for details).

## 5.2 IMAGENET DIT

Our study utilizes two pretrained teachers with the same DiT architecture: a standard flow matching (FM) DiT (the baseline in Chen et al. (2025)), and the REPA DiT (Yu et al., 2025). Interval CFG (Kynkäänniemi et al., 2024) is applied to both teachers to maximize their performance. Each $\pi$-Flow student is initialized with the teacher weights and then fully finetuned using the $\pi$-ID loss.

**Evaluation metrics.** We adopt the standard evaluation protocol in ADM (Dhariwal & Nichol, 2021) with the following metrics: Fréchet Inception Distance (FID) (Heusel et al., 2017), Inception Score (IS), and Precision–Recall (Kynkäänniemi et al., 2019).

**Comparison of DX and GMFlow policies.** As shown in Table 1, both policies yield strong 1-NFE FIDs after 40k training iterations, with the GMFlow policy consistently outperforming the DX policy by a clear margin. Notably, the DX policy exhibits sensitivity to the hyperparameter $N$ (number of grid points), whereas the GMFlow policy produces consistent results across different values of $K$ (number of Gaussians).

**Comparison with prior few-step DiTs.** In Table 2, we compare $\pi$-Flow (GM policy with $K = 32$) to prior few-step DiTs on ImageNet $256^2$: iCT (Song & Dhariwal, 2024), Shortcut models (Frans et al., 2025), iMM (Zhou et al., 2025), MeanFlow (Geng et al., 2025), and the concurrent work FACM (Peng et al., 2025). FACM (distilled from REPA) improves MeanFlow with an auxiliary loss and attains a leading 2-NFE FID, though it still relies on the inefficient JVP operation. In contrast, $\pi$-Flow uses a minimal training framework with no JVP and adaptive loss scalings, yet still outperforms the original MeanFlow DiT across both 1-NFE and 2-NFE generation.

**Ablation study on GM dropout.** From the two bottom rows in Table 1 we conclude that our standard implementation with a 0.05 GM dropout rate yields better FID and Recall compared to the setting without dropout, confirming the effectiveness of our GM dropout technique.

Table 3: Quantitative comparisons on COCO-10k dataset and HPSv2 prompt set.

| Model | Distill method | NFE | COCO-10k prompts | | | | | HPSv2 prompts | | | | |
|---|---|---|---|---|---|---|---|---|---|---|---|---|
| | | | Data align. | | Prompt align. | | Pref. align. | Teacher align. | | Prompt align. | | Pref. align. |
| | | | FID↓ | pFID↓ | CLIP↑ | VQA↑ | HPSv2.1↑ | FID↓ | pFID↓ | CLIP↑ | VQA↑ | HPSv2.1↑ |
| FLUX.1 dev | - | 50 | 27.8 | 34.9 | 0.268 | 0.900 | 0.309 | - | - | 0.284 | 0.805 | 0.314 |
| FLUX Turbo | GAN | 8 | **26.7** | **32.0** | 0.267 | 0.900 | 0.308 | 13.8 | 18.5 | **0.286** | **0.814** | 0.313 |
| Hyper-FLUX | CD+Re | 8 | 29.8 | 33.3 | **0.268** | 0.894 | 0.309 | 15.6 | 22.2 | 0.285 | 0.807 | 0.315 |
| π-Flow (GM-FLUX) | π-ID | 8 | 29.0 | 35.4 | **0.268** | **0.901** | **0.311** | **12.6** | **15.9** | 0.285 | 0.810 | **0.316** |
| SenseFlow (FLUX) | VSD+CD+GAN | 4 | 34.1 | 44.2 | 0.266 | 0.879 | 0.308 | 23.3 | 28.2 | 0.283 | 0.806 | **0.318** |
| π-Flow (GM-FLUX) | π-ID | 4 | 29.8 | **36.1** | 0.269 | 0.903 | 0.308 | **14.3** | **19.2** | **0.288** | **0.816** | 0.313 |
| π-Flow (GM-FLUX) | π-ID (data-free) | 4 | **29.7** | 36.2 | 0.269 | **0.905** | **0.310** | 14.4 | 19.7 | 0.287 | 0.813 | 0.314 |
| Qwen-Image | - | 50×2 | 34.1 | 45.6 | 0.282 | 0.936 | 0.312 | - | - | 0.302 | 0.872 | 0.309 |
| Qwen-Image Lightning | VSD | 4 | 37.5 | 51.6 | 0.280 | 0.935 | **0.322** | 15.6 | 19.7 | 0.299 | **0.867** | **0.328** |
| π-Flow (GM-Qwen) | π-ID | 4 | **36.0** | 46.1 | 0.281 | 0.934 | 0.314 | **12.8** | **16.6** | 0.300 | 0.860 | 0.310 |
| π-Flow (GM-Qwen) | π-ID (data-free) | 4 | **36.0** | **45.7** | 0.282 | 0.936 | 0.315 | 12.9 | 16.8 | **0.301** | 0.862 | 0.312 |

Table 4: Quantitative comparisons on OneIG-Bench (Chang et al., 2025).

| Model | Distill Method | NFE | Alignment↑ | Text↑ | Diversity↑ | Style↑ | Reasoning↑ |
|---|---|---|---|---|---|---|---|
| FLUX.1 dev | - | 50 | 0.790 | 0.556 | 0.238 | 0.370 | 0.257 |
| FLUX Turbo | GAN | 8 | 0.791 | 0.334 | **0.234** | **0.370** | 0.239 |
| Hyper-FLUX | CD+Re | 8 | 0.790 | **0.530** | 0.198 | 0.369 | 0.254 |
| π-Flow (GM-FLUX) | π-ID | 8 | **0.792** | 0.517 | **0.234** | 0.369 | **0.256** |
| SenseFlow (FLUX) | VSD+CD+GAN | 4 | 0.776 | 0.384 | 0.151 | 0.343 | 0.238 |
| π-Flow (GM-FLUX) | π-ID | 4 | **0.799** | 0.437 | **0.229** | 0.360 | **0.251** |
| π-Flow (GM-FLUX) | π-ID (data-free) | 4 | **0.799** | **0.460** | 0.224 | **0.363** | 0.249 |
| Qwen-Image | - | 50×2 | 0.880 | 0.888 | 0.194 | 0.427 | 0.306 |
| Qwen-Image Lightning | VSD | 4 | **0.885** | **0.923** | 0.116 | 0.417 | **0.311** |
| π-Flow (GM-Qwen) | π-ID | 4 | 0.875 | 0.892 | **0.180** | **0.434** | 0.298 |
| π-Flow (GM-Qwen) | π-ID (data-free) | 4 | 0.881 | 0.890 | 0.176 | 0.433 | 0.300 |

## 5.3 FLUX.1-12B AND QWEN-IMAGE-20B

For text-to-image generation, we distill the 12B FLUX.1 dev (Black Forest Labs, 2024b) and 20B Qwen-Image (Wu et al., 2025) models into π-Flow students. During student training, we freeze the base parameters inherited from the teacher and finetune only the expanded output layer along with 256-rank LoRA adapters (Hu et al., 2022) on the feed-forward layers. For data-dependent distillation, we prepare 2.3M one-megapixel (1MP) images captioned with Qwen2.5-VL (Bai et al., 2025). In the data-free setting, we use only the generated captions as conditioning inputs while keeping the same 1MP resolution when initializing the noise.

**Evaluation protocol.** We conduct a comprehensive evaluation on $1024^2$ high-resolution image generation from three distinct prompt sets: (a) 10K captions from the COCO 2014 validation set (Lin et al., 2014), (b) 3200 prompts from the HPSv2 benchmark (Wu et al., 2023), and (c) 1120 prompts from OneIG-Bench (Chang et al., 2025). For the COCO and HPSv2 sets, we report common metrics including FID (Heusel et al., 2017), patch FID (pFID) (Lin et al., 2024a), CLIP similarity (Radford et al., 2021), VQAScore (Lin et al., 2024b), and HPSv2.1 (Wu et al., 2023). On COCO prompts, FIDs are computed against real images, reflecting data alignment. On HPSv2, FIDs are computed against the 50-step teacher generations, reflecting teacher alignment. CLIP and VQAScore measure prompt alignment, while HPSv2 captures human preference alignment. For OneIG-Bench, we adopt its official evaluation protocol and metrics. All quantitative results are presented in Table 3 and 4.

**Competitor models.** We compare π-Flow against other few-step student models distilled from the same teacher. For FLUX, we compare against: 4-NFE SenseFlow (Ge et al., 2026), primarily leveraging variational score distillation (VSD) (Wang et al., 2023), also known as distribution matching distillation (DMD) (Yin et al., 2024b); 8-NFE Hyper-FLUX (Ren et al., 2024), trained with consistency distillation (CD) (Song et al., 2023) and reward models (Re) (Xu et al., 2023); 8-NFE FLUX Turbo, based on GAN-like adversarial distillation (Goodfellow et al., 2014; Sauer et al., 2024b). For Qwen-Image, we compare with the 4-NFE Qwen-Image Lighting based on VSD (ModelTC, 2025). Note that the 4-NFE FLUX.1 schnell is distilled from the closed-source FLUX.1 pro instead of the

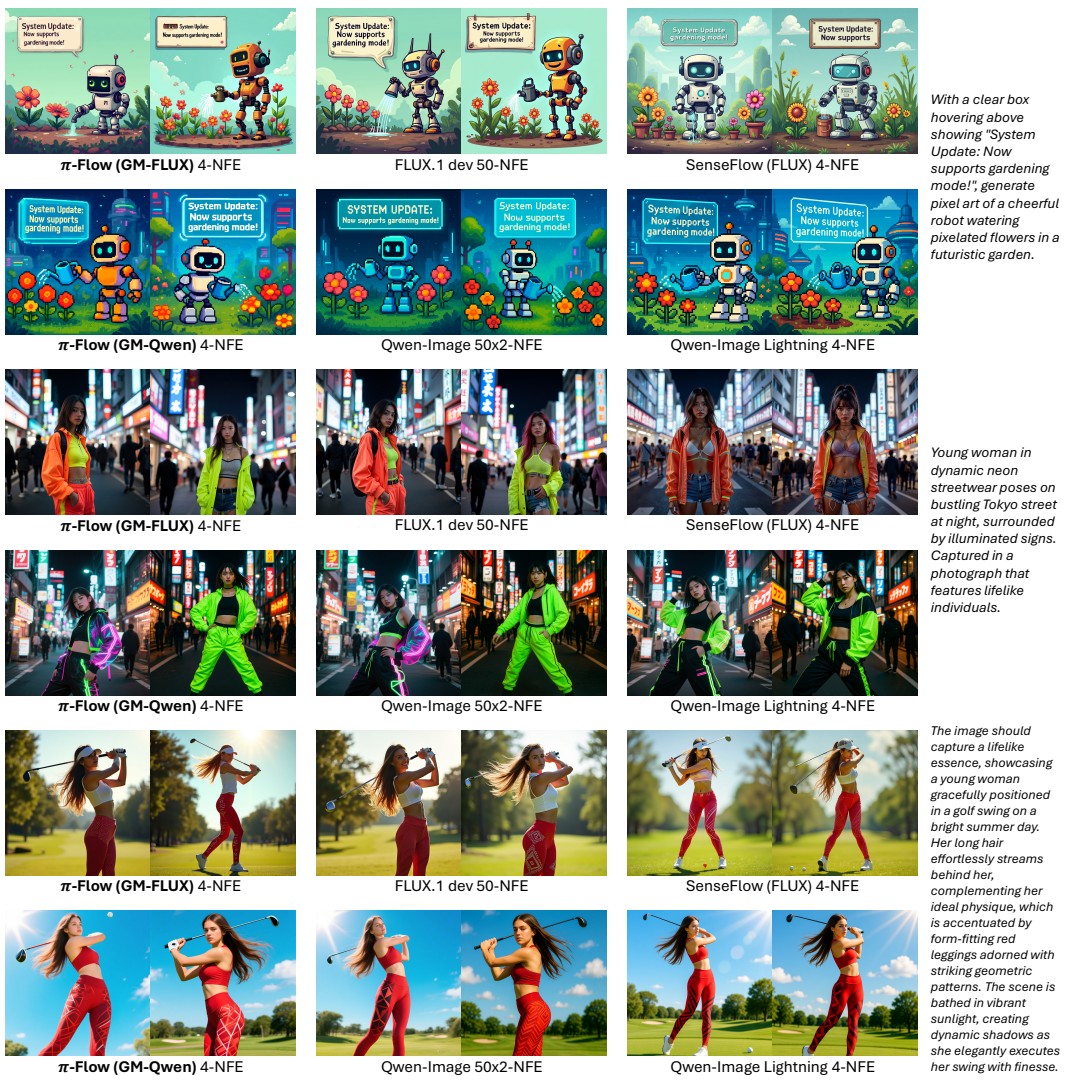

Figure 4: Images generated from the same batch of initial noise by $\pi$-Flows, teachers, and VSD students (SenseFlow, Qwen-Image Lightning). $\pi$-Flow models produce diverse structures that closely mirror the teacher's. In contrast, VSD students tend to repeat the same structure. Notably, Sense-Flow mostly generates symmetric images.

publicly available FLUX.1 dev (Black Forest Labs, 2024a), so we do not compare with it directly, but include further discussion in § D.

**Strong all-around performance.** As shown in Table 3 and Table 4, $\pi$-Flow demonstrates strong all-around performance, outperforming other few-step students on roughly 70% of all metrics, without exhibiting obvious weaknesses in any specific area.

**Superior diversity and teacher alignment.** $\pi$-Flow consistently achieves the highest diversity scores and the best teacher-referenced FIDs by clear margins, especially in the 4-NFE setting. These results strongly suggest that $\pi$-Flow effectively avoids both diversity collapse and style drift. As a result, most of its scores closely match those of the teacher, with some even slightly surpassing the teacher scores (e.g., prompt alignment and several Qwen-Image OneIG scores). Its strong teacher alignment is also evident in Fig. 4, where $\pi$-Flow generates structurally similar images to the teacher's from the same initial noise.

**Comparison with VSD (DMD) students.** VSD models are notable for high visual quality, sometimes surpassing teachers in quality and preference metrics. However, they are widely known to

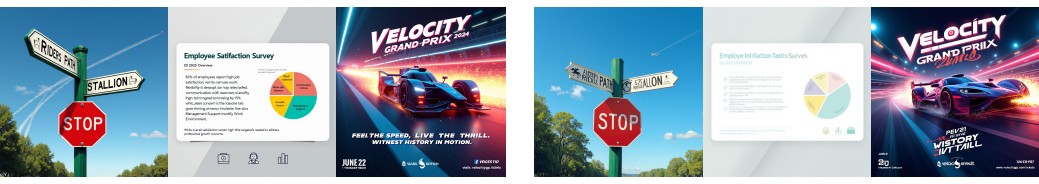

**π-Flow (GM-FLUX)** 8-NFE  FLUX Turbo 8-NFE

Figure 5: Images generated from the same initial noise by π-Flow and FLUX Turbo. π-Flow renders coherent texts, whereas FLUX Turbo underperforms in text rendering.

**π-Flow (GM-FLUX)** 8-NFE  Hyper-FLUX 8-NFE

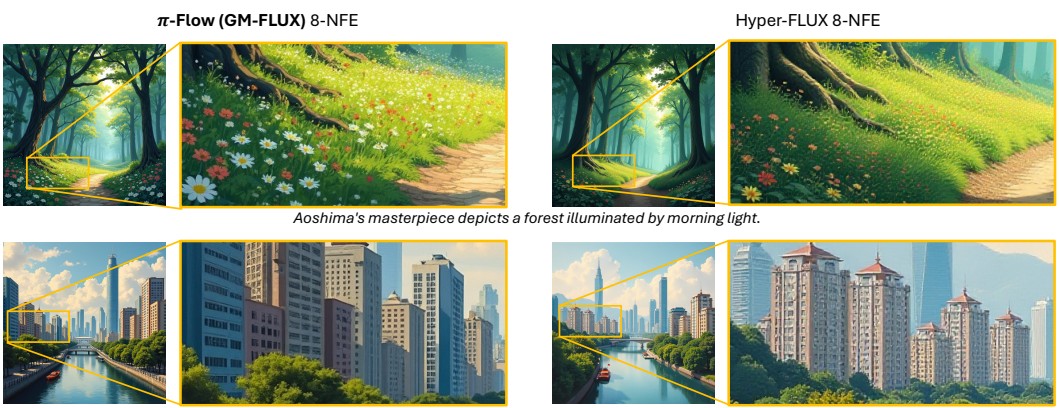

*Aoshima's masterpiece depicts a forest illuminated by morning light.*

*A painting depicting a scenic view of Guangzhou, China as a tourist destination by David Inshaw.*

Figure 6: Images generated from the same initial noise by π-Flow and Hyper-FLUX. π-Flow produces notably finer details, as highlighted in the zoomed-in patches.

suffer from mode collapse, as reflected in our experiments: both SenseFlow and Qwen-Image Lightning show significant drops in diversity and FIDs. Visual examples in Fig. 4 further highlight the collapse, where different initial noises produce visually similar images with only minor variations. In contrast, π-Flow maintains high quality and diversity without sacrificing either aspect.

**Comparison with other students.** FLUX Turbo achieves better data alignment FIDs than the teacher due to GAN training, yet its text rendering performance is significantly weaker, as shown in Fig. 5. Meanwhile, Hyper-FLUX often produces undesirable texture artifacts and fuzzy details, whereas π-Flow achieves superior detail rendering, as shown in Fig. 6.

**Data-dependent vs. data-free.** As shown in Table 3 and Table 4, data-dependent and data-free π-Flow models achieve nearly identical results. This demonstrates the practicality of π-Flow in scenarios where high-quality data is unavailable.

**GMFlow vs. DX policy.** Consistent with prior ImageNet findings, the DX policy slightly underperforms comapred to the GMFlow policy (Table 5), highlighting the latter's superior robustness.

**Convergence.** Figure 7 illustrates the convergence of π-Flow (GM-Qwen) over training iterations. Both FID and Patch FID scores initially improve rapidly, outperforming Qwen-Image Lightning within the first 400 iterations, and continue to improve steadily thereafter. This contrasts with previous GAN or VSD-based methods that often require frequent checkpointing and cherry-picking (Ge et al., 2026), demonstrating the scalability and robustness of our approach.

**Inference time.** To validate that the policies are indeed fast enough so that the overhead is negligible compared to shortcut-predicting models, we compare the inference times of 4-NFE π-Flow models and Qwen-Image Lightning in Table 6. For π-Flow, each policy generation step (with one network evaluation) is followed by 32 policy integration substeps on average. The results in Table 6 show that 32 policy substeps cost around 15 ms in total, which is only 3% of the network time. Therefore, the overall speed of π-Flow is on par with shortcut-predicting models.

Table 5: Comparisons between DX and GMFlow policies on text-to-image generation.

| Policy | Teacher | HPSv2 prompts | | | OneIG-Bench | |
|--------|---------|------|-------|---------|-------|-----------|
| | | FID↓ | pFID↓ | HPSv2.1↑ | Text↑ | Diversity↑ |
| DX ($N = 10$) | FLUX | 14.9 | 20.9 | **0.313** | 0.397 | 0.225 |
| GM ($K = 8$) | FLUX | **14.3** | **19.2** | **0.313** | **0.437** | **0.229** |
| DX ($N = 10$) | Qwen-Image | **12.7** | 17.0 | 0.306 | 0.869 | **0.185** |
| GM ($K = 8$) | Qwen-Image | 12.8 | **16.6** | **0.310** | **0.892** | 0.180 |

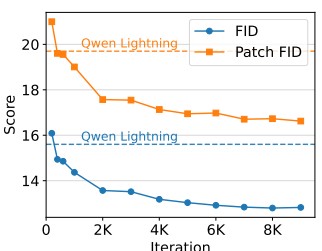

Figure 7: Teacher-referenced FID and Patch FID of GM-Qwen evaluated on HPSv2 prompts.

Table 6: Per-NFE inference time of $\pi$-Flow models and the shortcut-predicting model (Qwen-Image Lightning). Tested on an A100 GPU with 12 CPU cores (3.0 GHz).

| Model | Network time (sec) | Policy time (sec) |
|-------|--------------------|--------------------|
| Qwen-Image Lightning | 0.465 | - |
| $\pi$-Flow (DX-Qwen) | 0.464 | 0.015 |
| $\pi$-Flow (GM-Qwen) | 0.465 | 0.014 |

## 6 RELATED WORK

Prior work on diffusion model distillation primarily focuses on predicting shortcuts towards less noisy states, with training objectives ranging from direct regression to distribution matching.

Early work (Luhman & Luhman, 2021) directly regresses the teacher's ODE integral in a single step, but suffers from degraded quality, since regressing $x_0$ with an $\ell_2$ loss tends to produce blurry results. Progressive distillation methods (Salimans & Ho, 2022; Liu et al., 2023; 2024; Frans et al., 2025) make further improvements via a multi-stage process that progressively increases the student's step size and reduces its NFE by regressing the previous stage's multi-step outputs with fewer steps, yet this introduces error accumulation.

Consistency-based models (Song et al., 2023; Gu et al., 2023; Kim et al., 2024; Song & Dhariwal, 2024; Geng et al., 2025; Boffi et al., 2025) implicitly impose a velocity-based regression loss, which improves quality compared to $x$-based regression. However, the flow velocity of a shortcut-predicting student must be constructed implicitly using either inaccurate finite differences or expensive Jacobian–vector products (JVPs). Moreover, their quality is still limited due to accumulation of velocity errors into the integrated shortcut. Therefore, in practice, consistency distillation is often augmented with additional objectives to improve quality (Ren et al., 2024; Zheng et al., 2025), further complicating training.

Conversely, distribution matching approaches (Yin et al., 2024b;a; Sauer et al., 2024b; Zhou et al., 2024; Luo et al., 2024; Salimans et al., 2024; Zhou et al., 2025) adopt score matching and adversarial training to align the student's output distribution with the teacher's. The VSD objective achieves superior quality but risks diversity loss due to mode collapse; GAN and SiD objectives balance quality and diversity but can cause style drift. Their common reliance on auxiliary networks introduces additional tuning complexity and may lead to stability issues at scale (Ge et al., 2026).

## 7 CONCLUSION

We introduced policy-based flow models ($\pi$-Flow), a novel framework for few-step generation in which the network outputs a fast policy that enables accurate ODE integration via dense substeps reach the denoised state. To distill $\pi$-Flow models, we proposed a simple on-policy imitation learning approach that reduces the training objective to a single $\ell_2$ loss, mitigating error accumulation and quality–diversity trade-offs. Extensive experiments distilling ImageNet DiT, FLUX.1-12B, and Qwen-Image-20B models show that few-step $\pi$-Flows consistently attain teacher-level image quality while significantly outperforming competitors in diversity and teacher alignment. $\pi$-Flow offers a scalable, principled paradigm for efficient, high-quality generation and opens new directions for future research, such as exploring more robust policy families, improved distillation objectives, and extensions to other applications (e.g., video generation).

**Reproducibility statement.** To facilitate reproduction, we describe the detailed training procedures in Algorithms 2 and 3, and list all important hyperparameters in §C.

**Acknowledgements.** This project was partially done while Hansheng Chen was supported by the Qualcomm Innovation Fellowship and partially done while Hansheng Chen was an intern at Adobe Research. We would like to thank Jianming Zhang and Hailin Jin for their great support throughout the internship, and Xingtong Ge for the help in evaluating SenseFlow.

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

---

**Algorithm 2:** Data-dependent on-policy $\pi$-ID training loop with time shifting.

**Input:** $NFE$, teacher $G_{\boldsymbol{\theta}}$, data–condition distribution $p(\boldsymbol{x}_0, \boldsymbol{c})$, shift $m$
**Output:** Student $G_{\boldsymbol{\phi}}$

1   Initialize student params $\boldsymbol{\phi}$
2   $S \leftarrow \left\{ \frac{1}{NFE}, \frac{2}{NFE}, \cdots, 1 \right\}$             *// can be adjusted to reduce final step size*
3   **for** *finite samples* $\boldsymbol{x}_0, \boldsymbol{c} \sim p(\boldsymbol{x}_0, \boldsymbol{c})$*,* $\boldsymbol{\epsilon} \sim \mathcal{N}(\boldsymbol{0}, \boldsymbol{I})$*,* $\tau' \sim U(0,1)$ **do**
4      $\tau_{\mathrm{src}} \leftarrow \min\{ \tau_{\mathrm{src}} \mid \tau_{\mathrm{src}} \in S \text{ and } \tau_{\mathrm{src}} \geq \tau' \}$
5      $\tau_{\mathrm{dst}} \leftarrow \max\{ \tau_{\mathrm{dst}} \mid \tau_{\mathrm{dst}} \in S \cup \{ 0 \} \text{ and } \tau_{\mathrm{dst}} < \tau_{\mathrm{src}} \}$
6      $t_{\mathrm{src}} \leftarrow \frac{m\tau_{\mathrm{src}}}{1+(m-1)\tau_{\mathrm{src}}}$           *// time shifting (Esser et al., 2024)*
7      $\boldsymbol{x}_{t_{\mathrm{src}}} \leftarrow \alpha_{t_{\mathrm{src}}} \boldsymbol{x}_0 + \sigma_{t_{\mathrm{src}}} \boldsymbol{\epsilon}$
8      $\pi \leftarrow G_{\boldsymbol{\phi}}(\boldsymbol{x}_{t_{\mathrm{src}}}, t_{\mathrm{src}}, \boldsymbol{c})$
9      $\pi_{\mathrm{D}} \leftarrow \mathrm{stopgrad}(\pi) \text{ or } \pi_{\mathrm{D}} \leftarrow \mathrm{dropout}(\mathrm{stopgrad}(\pi))$
10     $\mathcal{L}_{\boldsymbol{\phi}} \leftarrow 0$
11     **for** *finite samples* $\tau \sim U(\tau_{\mathrm{dst}}, \tau_{\mathrm{src}})$ **do**
12        $t \leftarrow \frac{m\tau}{1+(m-1)\tau}$
13        $\boldsymbol{x}_t \leftarrow \boldsymbol{x}_{t_{\mathrm{src}}} + \int_{t_{\mathrm{src}}}^{t} \pi_{\mathrm{D}}(\boldsymbol{x}_t, t)\, \mathrm{d}t$
14        $\mathcal{L}_{\boldsymbol{\phi}} \leftarrow \mathcal{L}_{\boldsymbol{\phi}} + \frac{1}{2}\|G_{\boldsymbol{\theta}}(\boldsymbol{x}_t, t, \boldsymbol{c}) - \pi(\boldsymbol{x}_t, t)\|^2$      *// can be replaced with Eq. (6)*
15     $\boldsymbol{\phi} \leftarrow \mathrm{Adam}(\boldsymbol{\phi}, \nabla_{\boldsymbol{\phi}} \mathcal{L}_{\boldsymbol{\phi}})$               *// optimizer step*

---

**Algorithm 3:** Data-free on-policy $\pi$-ID training loop with time shifting.

**Input:** $NFE$, teacher $G_{\boldsymbol{\theta}}$, condition distribution $p(\boldsymbol{c})$, shift $m$
**Output:** Student $G_{\boldsymbol{\phi}}$

1   Initialize student params $\boldsymbol{\phi}$
2   **for** *finite samples* $\boldsymbol{c} \sim p(\boldsymbol{c})$*,* $\boldsymbol{x}_1 \sim \mathcal{N}(\boldsymbol{0}, \boldsymbol{I})$ **do**
3      $\tau_{\mathrm{src}} \leftarrow 1, \quad t_{\mathrm{src}} \leftarrow 1$
4      $\mathcal{L}_{\boldsymbol{\phi}} \leftarrow 0$
5      **while** $\tau_{\mathrm{src}} > 0$ **do**
6        $\tau_{\mathrm{dst}} \leftarrow \tau_{\mathrm{src}} - \frac{1}{NFE}$           *// can be adjusted to reduce final step size*
7        $t_{\mathrm{dst}} \leftarrow \frac{m\tau_{\mathrm{dst}}}{1+(m-1)\tau_{\mathrm{dst}}}$           *// time shifting (Esser et al., 2024)*
8        $\pi \leftarrow G_{\boldsymbol{\phi}}(\boldsymbol{x}_{t_{\mathrm{src}}}, t_{\mathrm{src}}, \boldsymbol{c})$
9        $\pi_{\mathrm{D}} \leftarrow \mathrm{stopgrad}(\pi) \text{ or } \pi_{\mathrm{D}} \leftarrow \mathrm{dropout}(\mathrm{stopgrad}(\pi))$
10       **for** *finite samples* $\tau \sim U(\tau_{\mathrm{dst}}, \tau_{\mathrm{src}})$ **do**
11          $t \leftarrow \frac{m\tau}{1+(m-1)\tau}$
12          $\boldsymbol{x}_t \leftarrow \boldsymbol{x}_{t_{\mathrm{src}}} + \int_{t_{\mathrm{src}}}^{t} \pi_{\mathrm{D}}(\boldsymbol{x}_t, t)\, \mathrm{d}t$
13          $\mathcal{L}_{\boldsymbol{\phi}} \leftarrow \mathcal{L}_{\boldsymbol{\phi}} + \frac{\tau_{\mathrm{src}} - \tau_{\mathrm{dst}}}{2}\|G_{\boldsymbol{\theta}}(\boldsymbol{x}_t, t, \boldsymbol{c}) - \pi(\boldsymbol{x}_t, t)\|^2$    *// can be replaced with Eq. (6)*
14        $\boldsymbol{x}_{t_{\mathrm{dst}}} \leftarrow \boldsymbol{x}_{t_{\mathrm{src}}} + \int_{t_{\mathrm{src}}}^{t_{\mathrm{dst}}} \pi_{\mathrm{D}}(\boldsymbol{x}_t, t)\, \mathrm{d}t$
15        $\tau_{\mathrm{src}} \leftarrow \tau_{\mathrm{dst}}, \quad t_{\mathrm{src}} \leftarrow t_{\mathrm{dst}}$
16      $\boldsymbol{\phi} \leftarrow \mathrm{Adam}(\boldsymbol{\phi}, \nabla_{\boldsymbol{\phi}} \mathcal{L}_{\boldsymbol{\phi}})$              *// optimizer step*

---

## A   USE OF LARGE LANGUAGE MODELS

In preparing this manuscript, we used large language models (LLMs) as general-purpose writing assistants for grammar corrections, rephrasing, and clarity/concision edits. All LLM-suggested edits were reviewed and verified by the authors, who take full responsibility for the final manuscript.

## B   ADDITIONAL TECHNICAL DETAILS

### B.1   GM TEMPERATURE

Inspired by the temperature parameter in language models, we introduce a similar temperature parameter for the GMFlow policy during inference. Let $T > 0$ be the temperature parameter. Given a $C$-dimensional GM velocity distribution $q(\boldsymbol{u}|\boldsymbol{x}_{t_{\mathrm{src}}}) = \sum_{k=1}^{K} A_k \mathcal{N}(\boldsymbol{u}; \boldsymbol{\mu}_k, s^2 \boldsymbol{I})$, the new GM prob-

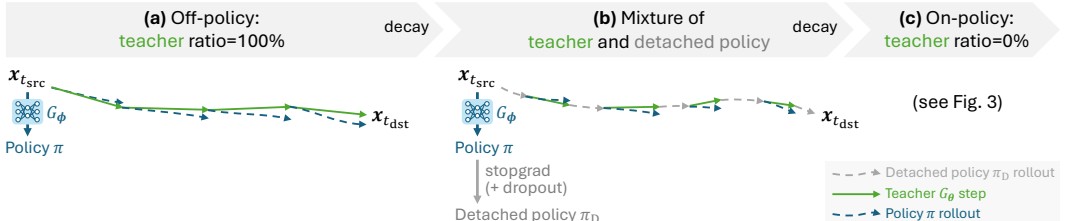

Figure 8: Three stages of scheduled trajectory mixing. (a) Off-policy behavior cloning with a teacher ratio of 1. (b) Mixed teacher and detached-policy segments with a decaying teacher ratio. (c) On-policy imitation learning with a teacher ratio of 0 (Fig. 3).

ability with temperature $T$ is defined as:

$$q_T(\boldsymbol{u}|\boldsymbol{x}_{t_{\text{src}}}) := \frac{q^{\frac{1}{T}}(\boldsymbol{u}|\boldsymbol{x}_{t_{\text{src}}})}{\int_{\mathbb{R}^C} q^{\frac{1}{T}}(\boldsymbol{u}|\boldsymbol{x}_{t_{\text{src}}})\,\mathrm{d}\boldsymbol{u}}. \tag{3}$$

Although $q_T(\boldsymbol{u}|\boldsymbol{x}_{t_{\text{src}}})$ does not have a general closed-form expression, it can be approximated by the following expression, which works very well as a practical implementation:

$$q_T(\boldsymbol{u}|\boldsymbol{x}_{t_{\text{src}}}) \approx \sum_{k=1}^{K} \frac{A_k^{\frac{1}{T}}}{\sum_{z=1}^{K} A_z^{\frac{1}{T}}} \mathcal{N}(\boldsymbol{u};\boldsymbol{\mu}_k, s^2 T\boldsymbol{I}). \tag{4}$$

For the distilled FLUX and Qwen-Image models, we set $T = 0.3$ for 4-NFE generation and $T = 0.7$ for 8-NFE generation. An exception is that we do not apply temperature scaling to the final step, as we found this can impair texture details. As shown in Table 7, ablating GM temperature from the 4-NFE GM-FLUX leads to degraded teacher alignment.

## B.2 SCHEDULED TRAJECTORY MIXING FOR GUIDANCE-DISTILLED TEACHERS

To reduce out-of-distribution exposure in imitation learning, scheduled sampling (Bengio et al., 2015) stochastically alternates between expert (teacher) and learner policy during trajectory integration, decaying the expert probability from 1 to 0. However, naively applying it to $\pi$-ID is impractical because the teacher flow model $G_{\boldsymbol{\theta}}$ is much slower than the network-free policy $\pi_{\text{D}}$.

To maintain constant compute throughout training, we introduce a scheduled trajectory mixing strategy. Since the teacher is slow, we fix the total number of teacher queries, allow each query to cover a coarse, longer step initially, and gradually shrink the teacher step size while filling the gaps with the fast policy $\pi_{\text{D}}$. As shown in Fig. 8 (a), training initially adopts a fully off-policy teacher trajectory (behavior cloning). At the beginning time $t_a$ of each teacher step, we roll in the learner policy $\pi$, integrate it over the same interval from $t_a$ to $t_b$, and match its average velocity to the teacher velocity with the $\ell_2$ loss:

$$\mathcal{L}_{\boldsymbol{\phi}} = \mathbb{E}\left[\frac{1}{2}\left\|G_{\boldsymbol{\theta}}(\boldsymbol{x}_{t_a}, t_a) - \frac{1}{t_b - t_a}\int_{t_a}^{t_b} \pi(\boldsymbol{x}_t, t)\,\mathrm{d}t\right\|^2\right]. \tag{5}$$

As training progresses (Fig. 8 (b)), we then mix teacher and detached-policy segments while using the same loss, and linearly decay the teacher ratio—the sum of teacher step lengths divided by the total interval length $t_{\text{src}} - t_{\text{dst}}$. Finally, when the teacher ratio reaches 0, training reduces to on-policy $\pi$-ID. All teacher step boundaries (starts and ends) are randomly sampled within the interval $[t_{\text{dst}}, t_{\text{src}}]$ under the teacher ratio constraint, so that step sizes and locations vary while the total teacher-covered length follows the current ratio schedule.

We apply scheduled trajectory mixing exclusively when distilling the FLUX.1 dev model, as it lacks real CFG. Since omitting CFG doubles the teacher's speed, we increase the number of intermediate samples (teacher steps) to 4 accordingly.

## B.3 MICRO-WINDOW VELOCITY MATCHING

For on-policy $\pi$-ID, in practice we found that replacing the instantaneous velocity matching loss in Algorithm 1 with a modified average velocity loss over a micro time window generally benefits

Table 7: Ablation study on 4-NFE $\pi$-Flow (GM-FLUX), evaluated on the HPSv2 prompt set using teacher-referenced FID metrics (reflecting teacher alignment).

| GM temperature | Micro window | FID↓ | pFID↓ |
|:---:|:---:|:---:|:---:|
| ✓ | ✓ | **14.3** | **19.2** |
| | ✓ | 14.9 | 20.1 |
| ✓ | | 14.6 | 20.3 |

FLUX.1 dev
43-NFE

FLUX.1 dev
128-NFE

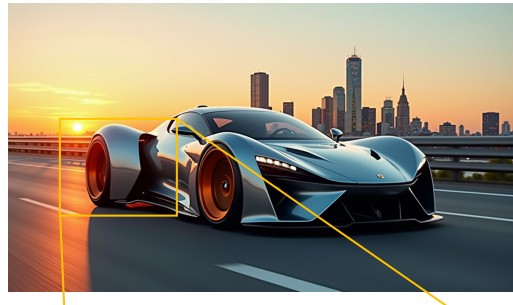
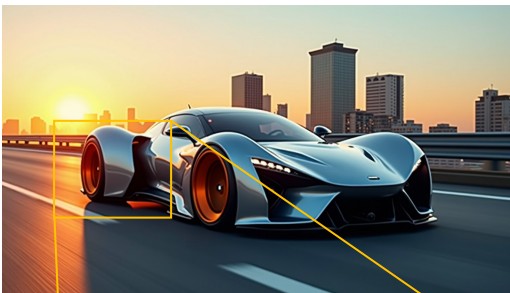

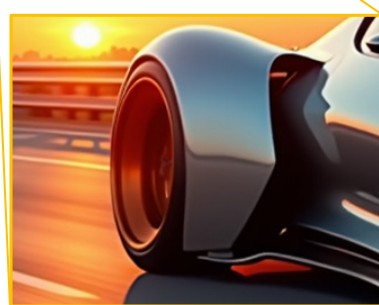
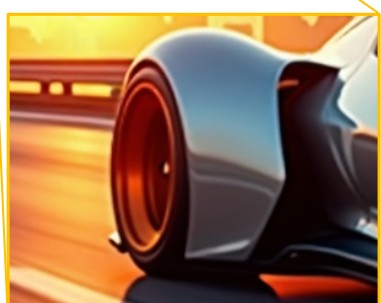

*A futuristic, sleek sports car with a low, aerodynamic design is shown in motion against a backdrop of a city skyline at sunset. The car features sharp angles, large wheels with orange accents, and a prominent front grille. The cityscape includes tall buildings with illuminated windows, and the sky is painted with hues of orange and blue as the sun sets. The lighting is warm and golden, with the sun setting behind the city, casting a glow over the scene. The car is positioned in the foreground, with the city skyline in the background, creating a sense of depth and movement.*

Figure 9: The 128-NFE FLUX.1 dev often generates blurry images, whereas the 43-NFE FLUX.1 dev reduces the blur and produces sharper edges.

training. The modified loss is defined as:

$$\mathcal{L}_\phi = \mathbb{E}\left[\frac{1}{2}\left\|G_{\boldsymbol{\theta}}(\boldsymbol{x}_t, t) - \frac{1}{-\Delta t}\int_t^{t-\Delta t}\pi(\boldsymbol{x}_t, t)\,\mathrm{d}t\right\|^2\right], \tag{6}$$

where $\Delta t$ is the window size. We set $\Delta t = 3/128$ (three policy integration steps) for all FLUX.1 and Qwen-Image experiments.

The benefits of micro-window velocity matching are threefold:

- It generally smooths the training signal, reducing sensitivity to sharp local variations in the teacher trajectory.
- It stabilizes the less robust DX policy. In the ImageNet experiments, we observe that training with the DX policy diverges without this modification.
- With $\Delta t = 3/128$, the policy effectively mimics teacher sampling with $\frac{128}{3} \approx 43$ steps instead of 128 steps. For the guidance-distilled FLUX.1 dev model, we observe that the teacher often generates blurry images using 128-step sampling, while 43-step sampling yields sharper results (see Fig. 9). This behavior is inherited by the student, so micro-window velocity matching helps reduce blur.

Table 8: Hyperparameters used in the ImageNet experiments.

| | 1-NFE | | | | | | 2-NFE |
|---|---|---|---|---|---|---|---|
| | GM-FM $(K=32)$ | GM-REPA $(K=8)$ | GM-REPA $(K=32)$ | DX-REPA $(N=10)$ | DX-REPA $(N=20)$ | DX-REPA $(N=40)$ | GM-REPA $(K=32)$ |
| GM dropout | 0.05 | 0.05 | 0.05 | - | - | - | 0.05 |
| # of intermediate states | 2 | 2 | 2 | 2 | 2 | 2 | 2 |
| Window size (raw) $\Delta\tau$ | - | - | - | 10/128 | 5/128 | 3/128 | - |
| Shift $m$ | 1.0 | 1.0 | 1.0 | 1.0 | 1.0 | 1.0 | 1.0 |
| Teacher CFG | 2.7 | 3.2 | 3.2 | 3.2 | 3.2 | 3.2 | 2.8 |
| Teacher CFG interval | $t \in [0, 0.6]$ | $t \in [0, 0.7]$ | $t \in [0, 0.7]$ | $t \in [0, 0.7]$ | $t \in [0, 0.7]$ | $t \in [0, 0.7]$ | $t \in [0, 0.7]$ |
| Learning rate | 5e-5 | 5e-5 | 5e-5 | 5e-5 | 5e-5 | 5e-5 | 5e-5 |
| Batch size | 4096 | 4096 | 4096 | 4096 | 4096 | 4096 | 4096 |
| # of training iterations in Table 2 | 140K | - | 140K | - | - | - | 24K |
| EMA param $\gamma$ in Karras et al. (2024) | 7.0 | 7.0 | 7.0 | 7.0 | 7.0 | 7.0 | 7.0 |

Table 9: Hyperparameters used in FLUX and Qwen-Image experiments.

| | 4-NFE | | | | 8-NFE |
|---|---|---|---|---|---|
| | GM-FLUX $(K=8)$ | GM-Qwen $(K=8)$ | DX-FLUX $(N=10)$ | DX-Qwen $(N=10)$ | GM-FLUX $(K=8)$ |
| GM dropout | 0.1 | 0.1 | - | - | 0.1 |
| GM temperature $T$ | 0.3 | 0.3 | - | - | 0.7 |
| # of intermediate states | 4 | 2 | 4 | 2 | 4 |
| Window size (raw) $\Delta\tau$ | 3/128 | 3/128 | 3/128 | 3/128 | 3/128 |
| Shift $m$ | 3.2 | 3.2 | 3.2 | 3.2 | 3.2 |
| Final step size scale | 0.5 | 0.5 | 0.5 | 0.5 | 0.5 |
| Teacher CFG | 3.5 | 4.0 | 3.5 | 4.0 | 3.5 |
| Learning rate | 1e-4 | 1e-4 | 1e-4 | 1e-4 | 1e-4 |
| Batch size | 256 | 256 | 256 | 256 | 256 |
| # of training iterations | 3K | 9K | 3K | 9K | 3K |
| # of decay iterations (§ B.2) | 2K | - | 2K | - | 2K |
| EMA param $\gamma$ in Karras et al. (2024) | 7.0 | 7.0 | 7.0 | 7.0 | 7.0 |

As shown in Table 7, ablating the micro window trick from the 4-NFE GM-FLUX leads to degraded teacher alignment.

## B.4 TIME SAMPLING

For high resolution image generation, Esser et al. (2024) proposed a time shifting mechanism to rescale the noise strength. Let $\tau$ be the pre-shift raw time and $m$ be the shift hyperparameter, the shifted time is defined as $t := \frac{m\tau}{1+(m-1)\tau}$.

Following this idea, $\pi$-ID samples times uniformly in raw-time space and then applies the shift to remap those samples. Detailed time sampling routines are given in Algorithms 2 and 3.

For FLUX.1 and Qwen-Image, we use a fixed shift $m = 3.2$, which is a rounded approximation of FLUX.1's official dynamic shift at 1MP resolution. In addition, several diffusion/flow models reduce the noise strength at the final step to improve detail (Karras et al., 2022; Wu et al., 2025). Accordingly, for FLUX.1 and Qwen-Image we halve the final step size (relative to previous steps) in raw-time space.

Table 10: FLUX.1 schnell evaluation results on COCO-10k dataset and HPSv2 prompt set.

| Model | Distill method | NFE | COCO-10k prompts | | | | | HPSv2 prompts | | |
| | | | Data align. | | Prompt align. | | Pref. align. | Prompt align. | | Pref. align. |
| | | | FID↓ | pFID↓ | CLIP↑ | VQA↑ | HPSv2.1↑ | CLIP↑ | VQA↑ | HPSv2.1↑ |
|-------|----------------|-----|------|-------|-------|------|---------|-------|------|---------|
| FLUX.1 schnell | GAN | 4 | 21.8 | 29.1 | 0.274 | 0.913 | 0.297 | 0.297 | 0.843 | 0.301 |

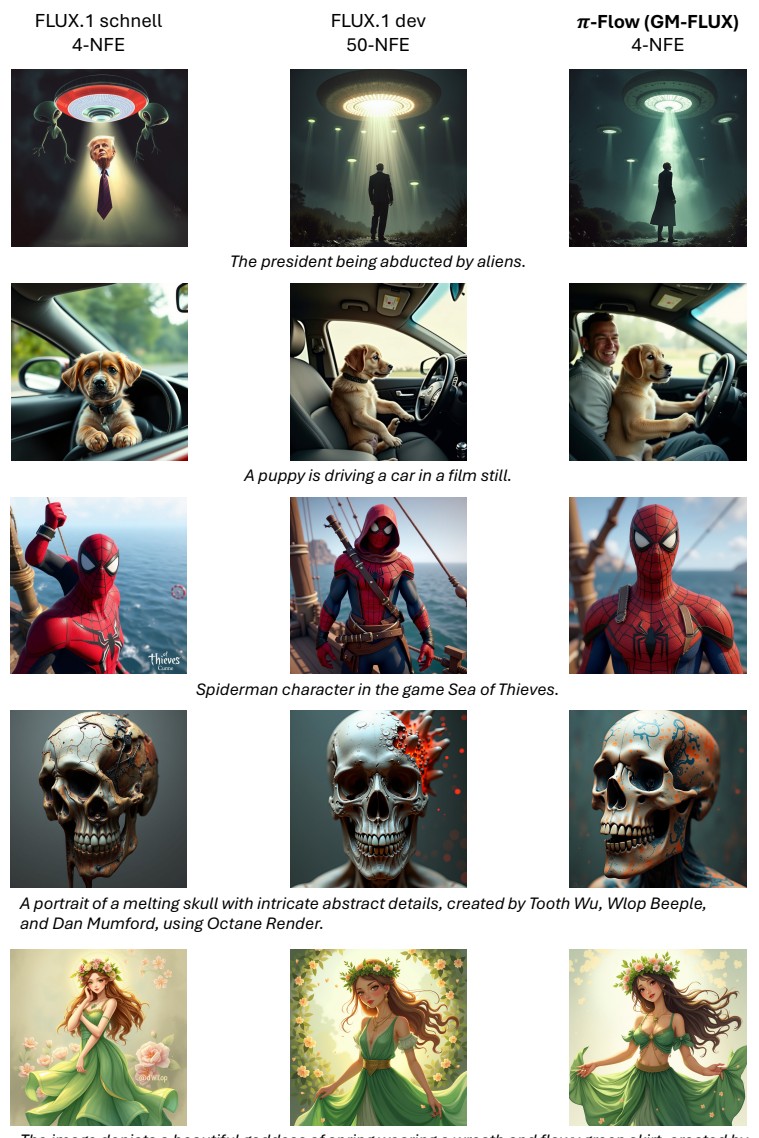

Figure 10: Typical failure cases of FLUX.1 schnell. For reference, we also show the corresponding FLUX.1 dev and π-Flow results from the same initial noise.

## C    ADDITIONAL IMPLEMENTATION DETAILS AND HYPERPARAMETERS

All models are trained with BF16 mixed precision, using the 8-bit Adam optimizer (Kingma & Ba, 2014; Dettmers et al., 2022) without weight decay. For inference, we use EMA weights with a dynamic moment schedule (Karras et al., 2024). Detailed hyperparameter choices are listed in Table 8 and 9.

Element-wise factorization        **Pixel-wise factorization**        Patch-wise factorization
$(H \times W \times 16) \times 1$        $(H \times W) \times 16$        $(H/2 \times W/2) \times 64$

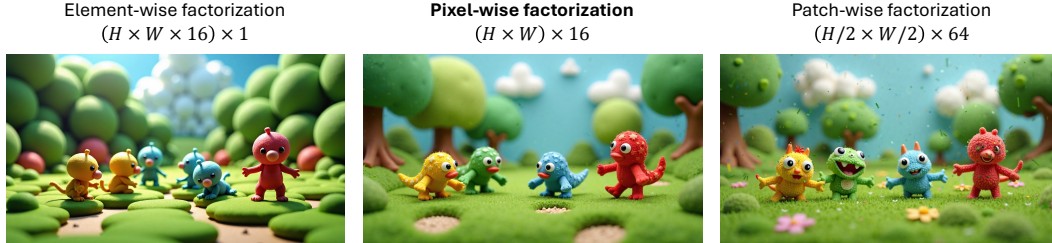

*Fuzzy clay creatures playing a game in a heavily tilted green pasture.*

Figure 11: Comparison of different $L \times C$ factorization schemes for the GMFlow policy, evaluated on the toy models after 4000 optimization iterations. The default pixel-wise factorization ($C =$ VAE latent channels) proposed by Chen et al. (2025) produces neutral colors and detailed textures. In contrast, element-wise factorization ($C = 1$) leads to over-saturated colors, high contrast, and over-smoothed textures, while patch-wise factorization ($C = 2 \times 2 \times$ VAE latent channels) results in "confetti" artifacts and noisy textures.

### C.1    DISCUSSION ON GMFLOW POLICY HYPERPARAMETERS

For the GMFlow policy, we observed that the hyperparameters suggested by Chen et al. (2025) ($K = 8, C =$ VAE latent channel size) generally work well. These parameters play important roles in balancing compatibility, expresiveness, and robustness. A larger $K$ improves expresiveness but impairs compatibility as it may complicate network training. A larger $C$ improves robustness (since GMFlow models correlations within each $C$-dimensional chunk) but impairs expresiveness (raises the theoretical $K = N \cdot C$ bound). In addition, improving expresiveness may generally compromise robustness, due to the increased chance of encountering outlier trajectories during inference.

To further justify the design choice of pixel-wise factorization in GMFlow (where the latent grid is factorized by $L = H \times W$, $C =$ VAE latent channel size), we conduct toy model experiments to test alternative factorizations. In each experiment, we initialize a set of GMFlow parameters $A_{ik}, \boldsymbol{\mu}_{ik}, s$ with $K = 32$ components, and directly optimize them to overfit the FLUX teacher's behavior over the entire time domain $t \in (0, 1]$, using simple $\pi$-ID (Algorithm 1) on a fixed initial noise $\boldsymbol{x}_1$ and a fixed text prompt. This setup isolates the inductive bias of the policy itself, without any influence from the student network. As shown in Fig. 11, pixel-wise factorization achieves the best results within 4000 optimization iterations, producing neutral colors and rich textures, and is therefore the most suitable choice for the GMFlow policy.

## D    DISCUSSION ON FLUX.1 SCHNELL

The official 4-NFE FLUX.1 schnell model (Black Forest Labs, 2024a) (based on adversarial distillation (Sauer et al., 2024a)) is distilled from the closed-source FLUX.1 pro instead of the publicly available FLUX.1 dev. This makes a direct comparison to the student models in Table 3 inequitable.

For reference, nevertheless, we include the COCO-10k and HPSv2 metrics for FLUX.1 schnell in Table 10. These metrics reveal a trade-off: while FLUX.1 schnell achieves significantly better data and prompt alignment than FLUX.1 dev, its preference alignment is substantially weaker than FLUX.1 dev and all of its students.

To validate this observation, we conducted a human preference study. Our 4-NFE $\pi$-Flow (GM-FLUX) was compared against FLUX.1 schnell on 200 images generated from HPSv2 prompts. $\pi$-Flow was preferred by users 59.5% of the time, aligning with the HPSv2.1 preference metric. Furthermore, qualitative comparisons in Fig. 10 reveals that FLUX.1 schnell is prone to frequent structural errors (e.g., missing/extra/distorted limbs), whereas $\pi$-Flow maintains coherent structures.

## E    PROOF OF THEOREM 1

We will prove that a GM with $N \cdot C$ components suffices for approximating any $N$-step trajectory in $\mathbb{R}^C$ by first establishing Theorem 2, and then applying the Richter–Tchakaloff theorem to show

that a mixture of $N \cdot C$ Dirac deltas satisfy all ODE moment equations, which finally leads to $N \cdot C$ Gaussian components.

**Theorem 2.** Given pairwise distinct times $t_1, \ldots, t_N \in (0, 1]$ and vectors $\boldsymbol{x}_{t_n}, \dot{\boldsymbol{x}}_{t_n} \in \mathbb{R}^C$ for $n = 1, \ldots, N$, there exists a probability measure $p(\mathrm{d}\boldsymbol{x}_0)$ on $\mathbb{R}^C$, such that Eq (1) holds at $t = t_n$ for every $n = 1, \ldots, N$.

### E.1 MOMENT EQUATION

For every $t \in (0, 1]$, the ODE moment equation has the following equivalent forms:

$$\dot{\boldsymbol{x}}_t = \int_{\mathbb{R}^C} \frac{\boldsymbol{x}_t - \boldsymbol{x}_0}{t} p(\mathrm{d}\boldsymbol{x}_0 | \boldsymbol{x}_t)$$

$$\Leftrightarrow \quad \dot{\boldsymbol{x}}_t \int_{\mathbb{R}^C} p(\mathrm{d}\boldsymbol{x}_0 | \boldsymbol{x}_t) = \int_{\mathbb{R}^C} \frac{\boldsymbol{x}_t - \boldsymbol{x}_0}{t} p(\mathrm{d}\boldsymbol{x}_0 | \boldsymbol{x}_t)$$

$$\Leftrightarrow \quad \int_{\mathbb{R}^C} \frac{\boldsymbol{x}_0 - \boldsymbol{x}_t + t\dot{\boldsymbol{x}}_t}{t} p(\mathrm{d}\boldsymbol{x}_0 | \boldsymbol{x}_t) = \boldsymbol{0}$$

$$\Leftrightarrow \quad \int_{\mathbb{R}^C} \frac{(\boldsymbol{x}_0 - \boldsymbol{x}_t + t\dot{\boldsymbol{x}}_t)\mathcal{N}\big(\boldsymbol{x}_t; \alpha_t \boldsymbol{x}_0, \sigma_t^2 \boldsymbol{I}\big) p(\mathrm{d}\boldsymbol{x}_0)}{t p(\boldsymbol{x}_t)} = \boldsymbol{0}$$

$$\Leftrightarrow \quad \int_{\mathbb{R}^C} (\boldsymbol{x}_0 - \boldsymbol{x}_t + t\dot{\boldsymbol{x}}_t)\mathcal{N}\big(\boldsymbol{x}_t; \alpha_t \boldsymbol{x}_0, \sigma_t^2 \boldsymbol{I}\big) p(\mathrm{d}\boldsymbol{x}_0) = \boldsymbol{0}. \tag{7}$$

Let $\boldsymbol{g}(t, \boldsymbol{x}_0) := (\boldsymbol{x}_0 - \boldsymbol{x}_t + t\dot{\boldsymbol{x}}_t)\mathcal{N}\big(\boldsymbol{x}_t; \alpha_t \boldsymbol{x}_0, \sigma_t^2 \boldsymbol{I}\big)$ be a kernel function. The above equation can be written as a multivariate homogeneous Fredholm integral equation of the first kind:

$$\int_{\mathbb{R}^C} \boldsymbol{g}(t, \boldsymbol{x}_0) p(\mathrm{d}\boldsymbol{x}_0) = \boldsymbol{0}. \tag{8}$$

To prove Theorem 2, we need to show that there exists a probability measure $p(\mathrm{d}\boldsymbol{x}_0)$ on $\mathbb{R}^C$ that solves the Fredholm equation at $t = t_n$ for every $n = 1, \cdots, N$.

### E.2 UNIVARIATE MOMENT EQUATION

To prove the existence of a solution to the multivariate Fredholm equation, we can simplify the proof into a univariate case by showing that an element-wise probability factorization $p(\mathrm{d}\boldsymbol{x}_0) = \prod_{i=1}^C p(\mathrm{d}x_{i0})$ exists that solves the Fredholm equation. In this case, Eq. (7) can be written as:

$$\forall i = 1, 2, \cdots, C,$$

$$\int_{\mathbb{R}} (x_{i0} - x_{it} + t\dot{x}_{it})\mathcal{N}\big(x_{it}; \alpha_t x_{i0}, \sigma_t^2\big) p(\mathrm{d}x_{i0}) \prod_{j \neq i} \int_{\mathbb{R}} \mathcal{N}\big(x_{jt}; \alpha_t x_{j0}, \sigma_t^2\big) p(\mathrm{d}x_{j0}) = 0$$

$$\Leftrightarrow \quad \forall i = 1, 2, \cdots, C, \quad \int_{\mathbb{R}} (x_{i0} - x_{it} + t\dot{x}_{it})\mathcal{N}\big(x_{it}; \alpha_t x_{i0}, \sigma_t^2\big) p(\mathrm{d}x_{i0}) = 0. \tag{9}$$

To see this, we need to prove that there exists a probability measure $p(x_0)$ on $\mathbb{R}$ that solves the following univariate Fredholm equation at $t = t_n$ for every $n = 1, \cdots, N$:

$$\int_{\mathbb{R}} g(t, x_0) p(\mathrm{d}x_0) = 0, \tag{10}$$

where $g(t, x_0) := (x_0 - x_t + t\dot{x}_t)\mathcal{N}\big(x_t; \alpha_t x_0, \sigma_t^2\big)$ is the univariate kernel function.

### E.3 CONVEX COMBINATION

**Lemma 1.** Define the vector function:

$$\boldsymbol{\gamma} \colon \mathbb{R} \to \mathbb{R}^N, \quad \boldsymbol{\gamma}(x_0) = (g(t_1, x_0), g(t_2, x_0), \cdots, g(t_N, x_0)). \tag{11}$$

Then, the zero vector lies in the convex hull in $\mathbb{R}^N$, i.e.:

$$\boldsymbol{0} \in \mathrm{conv}\{\,\boldsymbol{\gamma}(x_0) \mid x_0 \in \mathbb{R}\,\} \in \mathbb{R}^N. \tag{12}$$

*Proof.* Define $S := \mathrm{conv}\{\, \boldsymbol{\gamma}(x_0) \mid x_0 \in \mathbb{R} \,\}$. Assume for the sake of contradiction that $\mathbf{0} \notin S$.

By the supporting and separating hyperplane theorem, there exists $\boldsymbol{w} \neq \mathbf{0} \in \mathbb{R}^N$, such that:

$$\forall \boldsymbol{\chi} \in S, \quad \langle \boldsymbol{w}, \boldsymbol{\chi} \rangle \leq 0. \tag{13}$$

In particular, this implies that:

$$\forall x_0 \in \mathbb{R}, \quad \langle \boldsymbol{w}, \boldsymbol{\gamma}(x_0) \rangle \leq 0. \tag{14}$$

Define $h(x_0) := \langle \boldsymbol{w}, \boldsymbol{\gamma}(x_0) \rangle = \sum_{n=1}^{N} w_n g(t_n, x_0)$. Recall the definition of $g(t, x_0)$ that:

$$\begin{aligned} g(t, x_0) &= (x_0 - x_t + t\dot{x}_t)\mathcal{N}\big(x_t; \alpha_t x_0, \sigma_t^2\big) \\ &= \frac{x_0 - x_t + t\dot{x}_t}{\sqrt{2\pi t^2}} \exp\left(-\frac{(x_t - \alpha_t x_0)^2}{2\sigma_t^2}\right). \end{aligned} \tag{15}$$

Let $n^*$ be an index with $w_{n^*} \neq 0$ for which the exponential term above decays the slowest, i.e.:

$$\frac{\alpha_{t_{n^*}}^2}{2\sigma_{t_{n^*}}^2} = \min\left\{ \frac{\alpha_{t_n}^2}{2\sigma_{t_n}^2} \,\middle|\, w_{n^*} \neq 0 \right\}. \tag{16}$$

Note that since $\frac{\alpha_t^2}{2\sigma_t^2}$ is monotonic, for every $n \neq n^*$ with $w_n \neq 0$, we have $\frac{\alpha_{t_n}^2}{2\sigma_{t_n}^2} > \frac{\alpha_{t_{n^*}}^2}{2\sigma_{t_{n^*}}^2}$. Therefore, as $|x_0| \to \infty$, $h(x_0)$ is dominated by the $n^*$-th component, i.e.:

$$h(x_0) = w_{n^*} \frac{x_0 - x_{t_{n^*}} + t_{n^*}\dot{x}_{t_{n^*}}}{\sqrt{2\pi t_{n^*}^2}} \exp\left(-\frac{(x_{t_{n^*}} - \alpha_{t_{n^*}} x_0)^2}{2\sigma_{t_{n^*}}^2}\right)(1 + O(1)). \tag{17}$$

Because the term $x_0 - x_{t_{n^*}} + t_{n^*}\dot{x}_{t_{n^*}}$ changes sign between $-\infty$ and $+\infty$, $h(x_0)$ takes both positive and negative values. This contradicts the hyperplane implication that $h(x_0) \leq 0$. Therefore, we conclude that $\mathbf{0} \in S$. $\qquad\square$

By Lemma 1 and Carathéodory's theorem, the zero vector can be expressed as a convex combination of at most $N+1$ points on $\boldsymbol{\gamma}(x_0)$. Therefore, there exists a finite-support probability measure $p(\mathrm{d}x_0)$ consisting of $N + 1$ Dirac delta components that solves the univariate Fredholm equation at $t = t_n$ for every $n = 1, \cdots, N$, completing the proof of Theorem 2.

### E.4 $N \cdot C$ COMPONENTS SUFFICE

Richter's extension to Tchakaloff's theorem states as follows.

**Theorem 3** (Richter (1957); Tchakaloff (1957)). Let $\mathcal{V}$ be a finite-dimensional space of measurable functions on $\mathbb{R}^C$. For some probability measure $p(\mathrm{d}\boldsymbol{x}_0)$ on $\mathbb{R}^C$, define the moment functional:

$$\Lambda\colon \mathcal{V} \to \mathbb{R}, \quad \Lambda[g] := \int_{\mathbb{R}^C} g(\boldsymbol{x}_0) p(\mathrm{d}\boldsymbol{x}_0). \tag{18}$$

Then there exists a $K$-atomic measure $p^*(\mathrm{d}\boldsymbol{x}_0) = \sum_{k=1}^{K} A_k \delta_{\boldsymbol{\mu}_k}(\mathrm{d}\boldsymbol{x}_0)$ with $A_k > 0$ and $K \leq \dim \mathcal{V}$ such that:

$$\forall g \in \mathcal{V}, \quad \Lambda[g] = \int_{\mathbb{R}^C} g(\boldsymbol{x}_0) p^*(\mathrm{d}\boldsymbol{x}_0) = \sum_{k=1}^{K} A_k g(\boldsymbol{\mu}_k). \tag{19}$$

By Theorem 2, we know that for $\mathcal{V} = \mathrm{span}\{\, g_i(t_n, \boldsymbol{x}_0) \mid i = 1, \cdots, C, \ n = 1, \cdots, N \,\}$ with the scalar function $g_i(t_n, \boldsymbol{x}_0) := (x_{i0} - x_{it} + t\dot{x}_{it})\mathcal{N}\big(\boldsymbol{x}_t; \alpha_t \boldsymbol{x}_0, \sigma_t^2 \boldsymbol{I}\big)$, there exists a probability measure $p(\mathrm{d}\boldsymbol{x}_0)$ such that $\int_{\mathbb{R}^D} g_i(t_n, \boldsymbol{x}_0) p(\mathrm{d}\boldsymbol{x}_0) = 0$ for every $i, n$. Then, by the Richter–Tchakaloff theorem, there also exists a $K$-atomic measure with $K \leq \dim \mathcal{V} \leq N \cdot C$ that satisfies all the moment equations. By taking the upper bound, this implies the existence of an $N \cdot C$-atomic probability measure $p^*(\mathrm{d}\boldsymbol{x}_0) = \sum_{k=1}^{N \cdot C} A_k \delta_{\boldsymbol{\mu}_k}(\mathrm{d}\boldsymbol{x}_0)$ with $A_k > 0$, $\sum_{k=1}^{N \cdot C} A_k = 1$ that solves the Fredholm equation (Eq. (8)) at $t = t_n$ for every $n = 1, \cdots, N$.

Finally, since $\mathcal{N}\big(\boldsymbol{x}_t; \alpha_t \boldsymbol{x}_0, \sigma_t^2 \boldsymbol{I}\big)$ is continuous, the $N \cdot C$ Dirac deltas in $p^*(\mathrm{d}\boldsymbol{x}_0)$ can be replaced by a mixture of $N \cdot C$ narrow Gaussians, such that $\dot{\boldsymbol{x}}_n$ is approximated arbitrarily well for every $n$, i.e.:

$$
\begin{aligned}
&\forall n = 1, \cdots, N, \\
&\lim_{s \to 0} \frac{\boldsymbol{x}_{t_n} - \int_{\mathbb{R}^D} \boldsymbol{x}_0 p(\mathrm{d}\boldsymbol{x}_0 | \boldsymbol{x}_{t_n})}{t_n} \Bigg|_{p(\mathrm{d}\boldsymbol{x}_0) = \sum_{k=1}^{N \cdot C} A_k \mathcal{N}(\mathrm{d}\boldsymbol{x}_0; \boldsymbol{\mu}_k, s^2 I)} \\
&= \frac{\boldsymbol{x}_{t_n} - \int_{\mathbb{R}^D} \boldsymbol{x}_0 p(\mathrm{d}\boldsymbol{x}_0 | \boldsymbol{x}_{t_n})}{t_n} \Bigg|_{p(\mathrm{d}\boldsymbol{x}_0) = \sum_{k=1}^{N \cdot C} A_k \delta_{\boldsymbol{\mu}_k}(\mathrm{d}\boldsymbol{x}_0)} \\
&= \dot{\boldsymbol{x}}_{t_n}
\end{aligned}
\tag{20}
$$

This completes the proof of Theorem 1.

## F  DERIVATION OF CLOSED-FORM GMFLOW VELOCITY

In this section, we provide details regarding the derivation of closed-form GMFlow velocity, which was originally presented by Chen et al. (2025) but not covered in detail.

Given the $\boldsymbol{u}$-based GM prediction $q(\boldsymbol{u}|\boldsymbol{x}_{t_{\mathrm{src}}}) = \sum_{k=1}^{K} A_k \mathcal{N}\big(\boldsymbol{u}; \boldsymbol{\mu}_k, s^2 \boldsymbol{I}\big)$ with $A_k \in \mathbb{R}_+$, $\boldsymbol{\mu}_k \in \mathbb{R}^C$, $s \in \mathbb{R}_+$, we first convert it into the $\boldsymbol{x}_0$-based parameterization by substituting $\boldsymbol{u} = \frac{\boldsymbol{x}_{t_{\mathrm{src}}} - \boldsymbol{x}_0}{\sigma_{t_{\mathrm{src}}}}$ into the density function, which yields:

$$
q(\boldsymbol{x}_0|\boldsymbol{x}_{t_{\mathrm{src}}}) = \sum_{k=1}^{K} A_k \mathcal{N}\big(\boldsymbol{x}_0; \boldsymbol{\mu}_{\mathrm{x}k}, s_{\mathrm{x}}^2 \boldsymbol{I}\big),
\tag{21}
$$

with the new parameters $\boldsymbol{\mu}_{\mathrm{x}k} = \boldsymbol{x}_{t_{\mathrm{src}}} - \sigma_{t_{\mathrm{src}}} \boldsymbol{\mu}_k$ and $s_{\mathrm{x}} = \sigma_{t_{\mathrm{src}}} s$. Then, for any $t < t_{\mathrm{src}}$ and any $\boldsymbol{x}_t \in \mathbb{R}^C$, the denoising posterior at $(\boldsymbol{x}_t, t)$ is given by:

$$
q(\boldsymbol{x}_0|\boldsymbol{x}_t) = \frac{p(\boldsymbol{x}_t|\boldsymbol{x}_0)}{Z \cdot p(\boldsymbol{x}_{t_{\mathrm{src}}}|\boldsymbol{x}_0)} q(\boldsymbol{x}_0|\boldsymbol{x}_{t_{\mathrm{src}}}),
\tag{22}
$$

where $Z$ is a normalization factor dependent on $\boldsymbol{x}_{t_{\mathrm{src}}}, \boldsymbol{x}_t, t_{\mathrm{src}}, t$. Using the definition of forward diffusion $p(\boldsymbol{x}_t|\boldsymbol{x}_0) = \mathcal{N}\big(\boldsymbol{x}_t; \alpha_t \boldsymbol{x}_0, \sigma_t^2 \boldsymbol{I}\big)$, we have:

$$
\begin{aligned}
q(\boldsymbol{x}_0|\boldsymbol{x}_t) &= \frac{\mathcal{N}\big(\boldsymbol{x}_t; \alpha_t \boldsymbol{x}_0, \sigma_t^2 \boldsymbol{I}\big)}{Z \cdot \mathcal{N}\big(\boldsymbol{x}_{t_{\mathrm{src}}}; \alpha_{t_{\mathrm{src}}} \boldsymbol{x}_0, \sigma_{t_{\mathrm{src}}}^2 \boldsymbol{I}\big)} q(\boldsymbol{x}_0|\boldsymbol{x}_{t_{\mathrm{src}}}) \\
&= \frac{\mathcal{N}\big(\boldsymbol{x}_0; \frac{1}{\alpha_t} \boldsymbol{x}_t, \frac{\sigma_t^2}{\alpha_t^2} \boldsymbol{I}\big)}{Z' \cdot \mathcal{N}\big(\boldsymbol{x}_0; \frac{1}{\alpha_{t_{\mathrm{src}}}} \boldsymbol{x}_{t_{\mathrm{src}}}, \frac{\sigma_{t_{\mathrm{src}}}^2}{\alpha_{t_{\mathrm{src}}}^2} \boldsymbol{I}\big)} q(\boldsymbol{x}_0|\boldsymbol{x}_{t_{\mathrm{src}}}) \\
&= \frac{1}{Z''} \mathcal{N}\big(\boldsymbol{x}_0; \frac{\boldsymbol{\nu}}{\zeta}, \frac{\boldsymbol{I}}{\zeta}\big) q(\boldsymbol{x}_0|\boldsymbol{x}_{t_{\mathrm{src}}}), \\
&\quad \text{where} \quad \boldsymbol{\nu} = \frac{\alpha_t \boldsymbol{x}_t}{\sigma_t^2} - \frac{\alpha_{t_{\mathrm{src}}} \boldsymbol{x}_{t_{\mathrm{src}}}}{\sigma_{t_{\mathrm{src}}}^2}, \\
&\quad \phantom{\text{where} \quad} \zeta = \frac{\alpha_t^2}{\sigma_t^2} - \frac{\alpha_{t_{\mathrm{src}}}^2}{\sigma_{t_{\mathrm{src}}}^2}.
\end{aligned}
$$

The result can be further simplified into a new GM:

$$
q(\boldsymbol{x}_0|\boldsymbol{x}_t) = \sum_{k=1}^{K} A_k' \mathcal{N}\big(\boldsymbol{x}_0; \boldsymbol{\mu}_k', s'^2 \boldsymbol{I}\big),
\tag{23}
$$

with the following parameters:

$$s'^2 = \frac{s_\mathrm{x}^2}{s_\mathrm{x}^2 \zeta + 1} \tag{24}$$

$$\boldsymbol{\mu}'_k = \frac{s_\mathrm{x}^2 \boldsymbol{\nu} + \boldsymbol{\mu}_{\mathrm{x}k}}{s_\mathrm{x}^2 \zeta + 1} \tag{25}$$

$$A'_k = \frac{\exp a'_k}{\sum_{k=1}^{K} \exp a'_k}, \tag{26}$$

where the new logit $a'_k$ is given by:

$$a'_k = \log A_k + \frac{\boldsymbol{\mu}_{\mathrm{x}k} \cdot \left(\boldsymbol{\nu} - \frac{1}{2}\zeta \boldsymbol{\mu}_{\mathrm{x}k}\right)}{s_\mathrm{x}^2 \zeta + 1}. \tag{27}$$

Finally, the closed-form GMFlow velocity at $(\boldsymbol{x}_t, t)$ is given by function $\pi$:

$$\pi \colon \mathbb{R}^C \times \mathbb{R} \to \mathbb{R}^C, \quad \pi(\boldsymbol{x}_t, t) = \frac{\boldsymbol{x}_t - \mathbb{E}_{\boldsymbol{x}_0 \sim q(\boldsymbol{x}_0|\boldsymbol{x}_t)}[\boldsymbol{x}_0]}{t}$$

$$= \frac{\boldsymbol{x}_t - \sum_{k=1}^{K} A'_k \boldsymbol{\mu}'_k}{t}. \tag{28}$$

**Extension to discrete support.** The closed-form GMFlow velocity can also be generalized to discrete support by taking $\lim_{s_\mathrm{x} \to 0} \pi(\boldsymbol{x}_t, t)$, which yields the simplified parameters:

$$\boldsymbol{\mu}'_k = \boldsymbol{\mu}_{\mathrm{x}k} \tag{29}$$

$$a'_k = \log A_k + \boldsymbol{\mu}_{\mathrm{x}k} \cdot \left(\boldsymbol{\nu} - \frac{1}{2}\zeta \boldsymbol{\mu}_{\mathrm{x}k}\right). \tag{30}$$

## G    ADDITIONAL QUALITATIVE RESULTS.

We show additional uncurated results of FLUX-based models in Fig. 12 and 13.

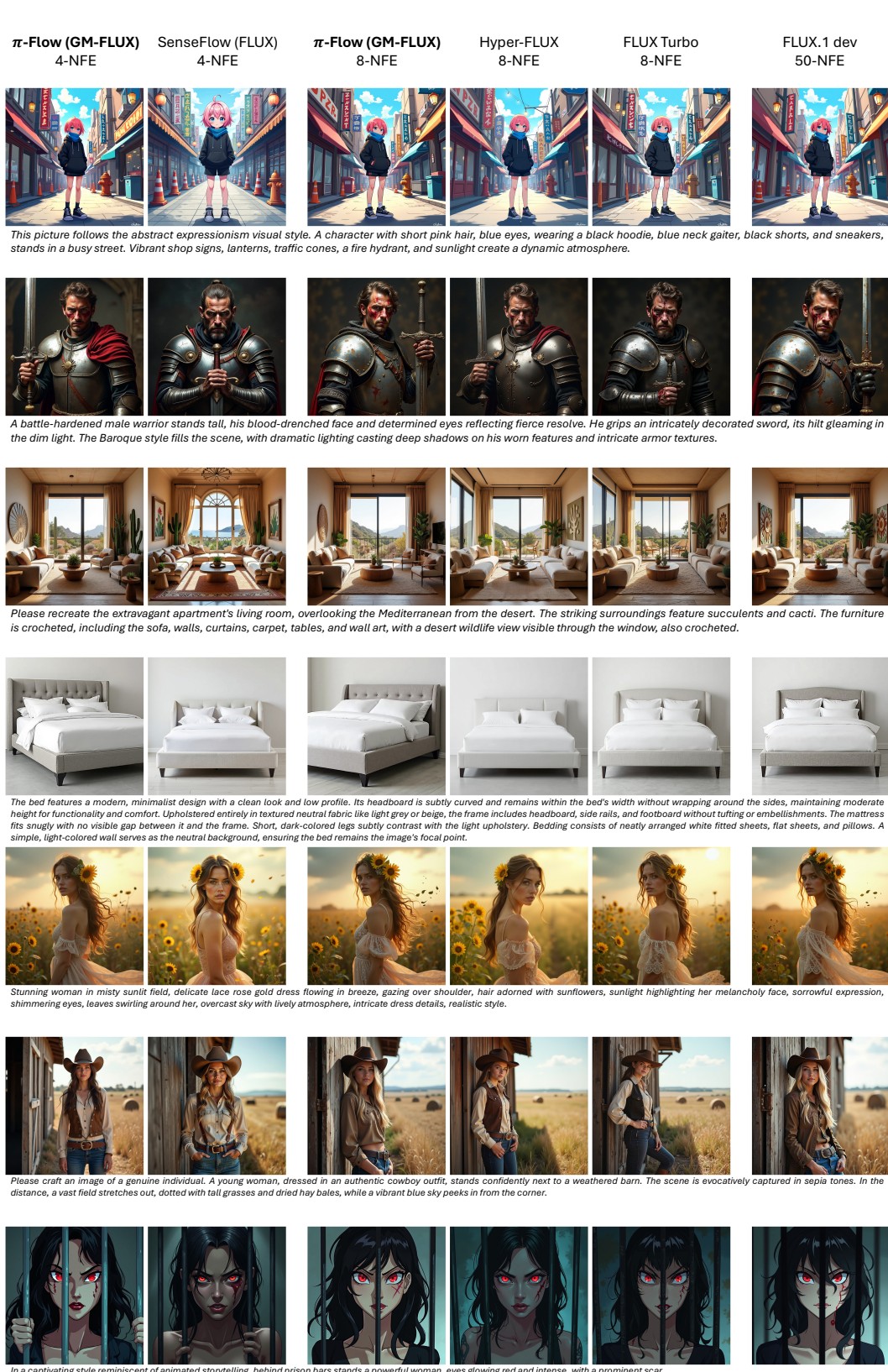

Figure 12: An uncurated random batch from the OneIG-Bench prompt set, part A.

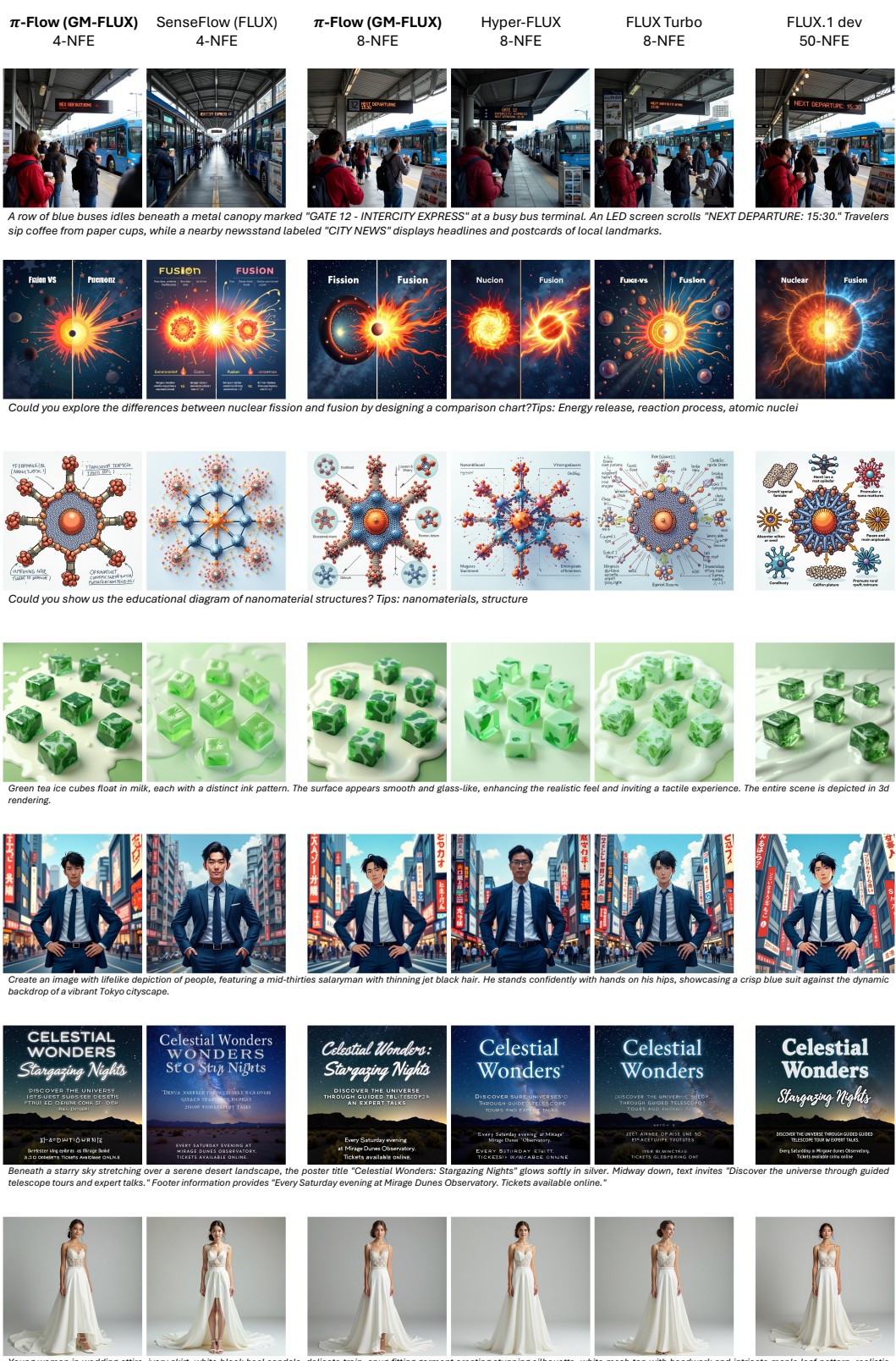

Figure 13: An uncurated random batch from the OneIG-Bench prompt set, part B.

