# OpenReview forum: "pi-Flow: Policy-Based Few-Step Generation via Imitation Distillation"
_ICLR.cc/2026/Conference — ICLR 2026 Poster_

### Official Review · Reviewer_1tXL · 2025-10-28

**Soundness:** 3
**Presentation:** 4
**Contribution:** 3
**Rating:** 8
**Confidence:** 4

**Summary:**

This paper describes a method for the distillation of diffusion/flow networks. It assumes a pretrained flow model, and aims to learn a student model, initialized from the same weights, that can produce outputs with a small number of function evaluations. The key novelty with relation to prior distillation methods is the idea to predict a *policy* rather than a direct velocity. This policy can then be used to perform ODE denoising over many iterations, without the need for further network calls. The two policy classes are a DX policy, which predicts a sequence of (x0) predictions according to various t, and a GMFlow policy, which predicts a Gaussian mixture over the data. The GMFlow policy can be solved analytically to predict the velocity at a given t. To train the student model, on-policy distillation is used to fit the policy's predictions to the teacher. The proposed method outperforms prior works on a range of benchmarks such as DiT, FLUX, and Qwen-Image distillation.

**Strengths:**

This paper provides a novel contribution in the idea to predict a "policy" from the student, rather than directly predicting velocity. This innovation allows a single network call to define a series of ODE steps, which is key to retaining performance while reducing the number of function calls. In terms of quality and clarity, the paper is well-written, with a clear explanation of the method and thorough experimental results. This paper has a chance to be significant in the sense of a practical, effective flow distillation method.

**Weaknesses:**

- The description of the GMFlow policy class is hard to understand, and could benefit from a more intuitive explanation. Specifically, the paper would benefit from a more explicit instantiation of the mapping from (student network) -> (policy parameters) -> (velocity prediction at x_t, t).
- The proposed method requires denoising the policy via an ODE *during the training process*, which adds a considerable overhead to the training pipeline. This stands in contrast to prior works which reduce the need for ODE rollouts during training (e.g. consistency models, progressive distillation, etc).
- The comparison in Table 2 is not necessarily fair, as it compares the proposed pi-Flow method, which is a distillation method that requires a pretrained teacher, to iCT/Shortcut/Meanflow which are *from-scratch* methods that do not require a teacher.

**Questions:**

- I am a bit confused how this distillation procedure avoids the mean-collapse issue with naive distillation. For example, if the fixed trajectory is at t=1 (i.e. pure noise), then the DX policy head will attempt to predict E[x0], which is just the mean under the dataset. Thus, how is it that the student network can be used to generate images in a single NFE (which by definition should map pure noise to an image?)
- How are t_src and t_dst chosen (page 5)? Does NFE stand for NFE under the teacher model, or under the student model?

---

> ### Author Response · Authors · 2025-11-28
>
> We thank the reviewer for the positive comments on novelty, writing quality and strong empirical results.
>
> **Intuitive explanation of GMFlow Policy**
>
> We thank the reviewer for the suggestions. We have updated the paper according to the suggestions (L197-199):
> > Intuitively, the student network $G_\phi$ maps the initial state $x_{t_\text{src}}$ to multiple denoising modes that parameterize the GMFlow policy. The policy then enables a closed-form velocity expression at future state $(x_t, t)$ for any $0 < t < t_\text{src}$.
>
> **Training overhead**
>
> We’d like to clarify that pi-Flow is highly efficient in training (in terms of training time per student update iteration) when compared to other distillation frameworks.
>
> Particularly, policy-based ODE rollout has negligible overhead when compared to network evaluations (only 3% extra time per network evaluation), as reported in the updated paper (Table 6 in page 10).
>
> Thus, pi-Flow training effectively needs only one forward and one backward pass of the student. In contrast, JVP-based consistency distillation (e.g. MeanFlow) requires an additional JVP forward pass of the student (in our response to Reviewer 3qdq, we showed that the wall time of pi-Flow is roughly half of the JVP-based concurrent work FACM), and VSD (DMD) methods require additional fake score forward and backward passes.
>
> **Comparisons in Table 2**
>
> We would like to note that we have demonstrated pi-Flow’s effectiveness as a distillation framework in the large-scale text-to-image setting (Table 3, Table 4), which we believe is a more practical use case: pi-Flow outperforms SOTA VSD (DMD) methods, as well as other methods based on GAN and consistency distillation (CD).
>
> On ImageNet, we have already compared pi-Flow with the concurrent distillation method FACM for completeness in Table 2.
>
> **How does this distillation procedure avoid the mean-collapse issue**
>
> Great question! DX policy head is not just predicting $E[x_0|x_1]$, but a grid of $E[x_0|x_{t_i}]$ for multiple times $t_1, \dots, t_N \in [t_\text{dst}, t_\text{src}]$. This is possible because all the intermediate states $x_{t_i}$ are deterministic based on the initial state $x_1$ and the teacher ODE, so the posterior mean $E[x_0|x_{t_i}]$ is also deterministic and can all be inferred from the initial state $x_1$.
>
> From a high-level perspective, both DX and GMFlow policy classes are expressive enough to represent arbitrary ODE trajectories as discussed in Section 3. In the updated paper (L251-256), we added a discussion on the convergence behavior of pi-ID: same as DAgger, it converges to the best policy in the function class, under the student’s capacity constraint.
>
> **How are t_src and t_dst chosen? Does NFE stand for NFE under the teacher model, or under the student model?**
>
>  Details on time sampling are provided in the appendix (Section B.4, Algorithm 2 and 3). In short, we divide the entire time interval $[0, 1]$ into consecutive segments according to student NFEs. In the ImageNet experiment, for example, the 2-NFE student has two evenly segmented time intervals $[0.5, 1.0]$ and $[0.0, 0.5]$, corresponding to $[t_\text{dst}, t_\text{src}]$. For large text-to-image models, these segments are not equal in length due to the time shifting mechanism.

---

### Official Review · Reviewer_EmE8 · 2025-10-30

**Soundness:** 3
**Presentation:** 3
**Contribution:** 3
**Rating:** 4
**Confidence:** 4

**Summary:**

This paper introduces Pi-Flow, a novel framework for distilling fast samplers by training a policy model rather than a traditional trajectory-based generator. Instead of predicting instantaneous velocities along a diffusion trajectory, Pi-Flow outputs an entire policy function that governs the trajectory’s evolution. This approach allows the distilled model to represent the generative process as a global functional mapping, rather than a sequence of stepwise updates.

**Strengths:**

1. **Novel Policy-Based Distillation Approach**
   Distilling a policy function that predicts an entire trajectory, while conceptually related to operator learning in DFNO (Deep Fourier Neural Operator) [1], is novel and interesting.

2. **Comprehensive Large-Scale Experiments**
   The authors conduct extensive experiments on large-scale text-to-image models such as **Flux** and **Qwen**, demonstrating that the method scales effectively to state-of-the-art architectures.

3. **Strong Text-to-Image Performance**
   Empirical results show that Pi-Flow outperforms prior distillation methods in the text-to-image domain, highlighting its potential as a scalable and efficient alternative to conventional diffusion distillation techniques.

**Weaknesses:**

1. **Unfair Baseline Comparisons**
   In Table 2, the paper primarily compares Pi-Flow with training-from-scratch methods such as iMM and Mean Flow, rather than distillation-based methods like Score Identity Distillation (SiD) [2] or Consistency Training Models (CTM) [3]. This makes it difficult to assess Pi-Flow’s relative effectiveness as a distillation framework.

2. **Limited Evaluation on Standard Diffusion Backbones**
   While large-scale experiments on Flux demonstrate the method’s scalability, evaluations on more established diffusion models such as **Stable Diffusion 1.5** and **SDXL** would strengthen the paper. These benchmarks are widely used in prior distillation works (e.g., Long and Short Classifier-Free Guidance [4], Phased Consistency Models [5]), and results on them would allow for a fairer comparison.

3. **Inconsistent Efficiency Reporting**
   Since Pi-Flow predicts the full trajectory at once (i.e., the entire policy function), the architecture requires increased channel capacity and computational cost. Consequently, NFE (number of function evaluations) is no longer a fair efficiency metric. Metrics such as FLOPs or latency should be reported instead to provide a meaningful assessment of inference efficiency.

**Questions:**

1. **Clarify Multistep Training Scheme**
   The paper mentions both single-step and multistep training setups, but the procedure for multistep training is not clearly described. Additional details on how the multistep policy is trained, would improve clarity.

2. **Expand Description of GM Flow**
   The explanation of GM Flow is incomplete. The paper lacks a formal definition of the training objective for each policy and does not give sufficient details on how the architecture is adapted to enable policy generation. Providing explicit loss formulations, architectural modifications, and training details would make the contribution clearer and more reproducible.

---

## References
[1] **Fast Sampling of Diffusion Models via Operator Learning.**
[2] **Score identity Distillation: Exponentially Fast Distillation of Pretrained Diffusion Models for One-Step Generation.**
[3] **Consistency Trajectory Models: Learning Probability Flow ODE Trajectory of Diffusion.**
[4] **Guided Score identity Distillation for Data-Free One-Step Text-to-Image Generation.**
[5] **Phased Consistency Models.**

---

> ### Author Response · Authors · 2025-11-28
>
> We thank the reviewer for the positive comments on novelty and strong empirical results.
>
> **Baseline Comparisons in Table 2. Relative effectiveness as a distillation framework.**
>
> We would like to emphasize that we have demonstrated pi-Flow’s effectiveness as a distillation framework in the large-scale text-to-image setting (Table 3, Table 4), which we believe is a more practical use case: pi-Flow outperforms SOTA VSD (DMD) methods, as well as other methods based on GAN and consistency distillation (CD). Related works [2][3] have shown that other methods like SiD and standalone JVP-based method (without VSD) generally underperform in quality when compared to VSD.
>
> On ImageNet, we have compared pi-Flow with the concurrent *distillation* method FACM for completeness in Table 2. SiD and CTM only present ImageNet 64 results, based on pixel-space $\epsilon$-predicting diffusion models. Adapting them to flow-based latent models on ImageNet 256 is non-trivial: CTM is trained with LPIPS loss which is not applicable to latent flow models; we have tried to adapt SiD for the ImageNet 256 setting but come across stability issues in our experiments (score-based methods are widely known to be sometimes unstable).
>
> **Evaluation on “Standard Diffusion Backbones” (SD1.5 and SDXL)**
>
> In our experiments, we focus on recent state-of-the-art flow-matching models built on large-scale transformer architectures for text-to-image generation, including Flux and Qwen-Image. These architectures have become the de facto standard backbones for modern generative models. In contrast, SD1.5 and SDXL represent an earlier generation of $\epsilon$-predicting UNets and are therefore less relevant to our study.
>
> We have made comparisons to multiple VSD/GAN/CD-based baselines, including SenseFlow (arXiv 2025) and Hyper-SD (NeurIPS 2024) that have state-of-the-art performance on both Flux and SDXL. Prior distillation methods such as Phased Consistency Models have been demonstrated to fall short of these baselines (see Table 1 and Table 2 in [4]).
>
> **Efficiency Report**
>
> We added an inference time report in the updated paper (Table 6 in page 10). According to the results, the overall overhead of pi-Flow compared to other few-step methods is only 3%:
>  - Policy integration (128 total substeps) only introduces 3% overhead compared to the network evaluation cost (4 NFEs).
>  - Expanding the network output channel does not show measurable differences in time, as both pi-Qwen and Qwen-Image Lightning spend around 0.465 sec for each network evaluation.
>  Therefore, as described in the paper, pi-Flow introduces negligible overhead compared to shortcut-predicting models, making NFE a fair efficiency metric.
>
> **Multistep Training Scheme**
>
> The complete multistep training scheme is described in the appendix (Algorithm 2 and 3). In short, multistep training performs imitation distillation on time intervals from $t_\text{src}$ to $t_\text{dst}$ (segments that each NFE covers). The difference between data-dependent and data-free multistep distillation lies in how the initial state $x_{t_\text{src}}$ is obtained: data-dependent distillation adds noise to the real data $x_0$ to obtain $x_{t_\text{src}}$, whereas data-free distillation uses the terminal state $x_{t_\text{dst}}$ of the previous segment as the initial state $x_{t_\text{src}}$ of the next segment.
>
> **Description of GM Flow. Training objective for each policy. How the architecture is adapted.**
>
> GMFlow is not a novel contribution in this paper. It has been extensively described in the cited paper [1]. In our appendix (§ F), we have further included details of the GMFlow velocity.
>
> The training objective of pi-ID is a standard L2 flow matching loss between the teacher velocity and policy velocity along the policy trajectory. Such an objective is policy-agnostic, as described in Algorithm 1. All policy classes are plug-and-play modules.
>
>  In § 3.1, 3.2, we have highlighted that architectures are adapted by expanding their output channels. Specifically, DX policy expands the output channels to predict a grid of $x_0^{(t_i)}$ values (defined as $E[x_0 | x_{t_i}]$) at N evenly spaced times $t_1, \dots, t_N \in [t_\text{dst}, t_\text{src}]$, so the output channels are N-times of the original head. GMFlow’s architecture adaptation has been extensively described in [1].
>
> **We hope that our replies have addressed the questions from the reviewer, and we kindly ask the reviewer to consider raising the score. We are happy to make further clarifications if the reviewer has more questions.**
>
>  [1] Chen et al. Gaussian Mixture Flow Matching Models. ICML 2025.
>
>  [2] Huang et al. Self Forcing: Bridging the Train-Test Gap in Autoregressive Video Diffusion. NeurIPS 2025.
>
>  [3] Zheng et al. Large Scale Diffusion Distillation via Score-Regularized Continuous-Time Consistency. arXiv 2025.
>
>  [4] Ge et al. SenseFlow: Scaling Distribution Matching for Flow-based Text-to-Image Distillation. arXiv 2025.

---

### Official Review · Reviewer_KQWK · 2025-10-31

**Soundness:** 3
**Presentation:** 3
**Contribution:** 3
**Rating:** 4
**Confidence:** 3

**Summary:**

This paper proposes pi-Flow, a new framework for few-step generative modeling that reframes flow-based distillation as policy learning. Instead of directly predicting the terminal point of a probability flow (as in velocity or consistency distillation), pi-Flow trains a network to output a policy that governs the velocity field across multiple integration substeps without additional network evaluations. The authors introduce a training algorithm, Policy-Based Imitation Distillation (pi-ID), inspired by on-policy imitation learning (DAgger), which lets the student model imitate the teacher’s flow dynamics along its own trajectory. Two specific parameterizations are explored:DX and GMFlow. Empirical results on ImageNet-256 and large text-to-image backbones (FLUX, Qwen-Image) show that pi-Flow achieves superior quality–diversity trade-offs and state-of-the-art one-step and few-step sampling performance.

**Strengths:**

1. The policy-based perspective is a conceptually elegant and novel reinterpretation of few-step distillation, bridging diffusion/flow modeling and imitation learning.
2. The pi-ID training procedure is well-motivated and effectively stabilizes learning by mitigating compounding errors—an issue that plagues prior consistency and flow distillation methods.
3. The proposed network-free policy formulation enables fine-grained ODE integration with very few network calls, yielding strong efficiency gains.

**Weaknesses:**

1. While the policy formulation is innovative, the theoretical justification (why and when a policy yields better approximation of teacher trajectories) is mostly intuitive; a more formal analysis of error bounds or convergence would strengthen the contribution.
2. The DX and GMFlow variants differ significantly in nature, but the paper does not fully disentangle whether gains arise from the pi-Flow framework itself or the added modeling flexibility of GMFlow.
3. The approach introduces additional hyperparameters (policy length, Gaussian components) whose tuning cost is not well quantified.
4. Although pi-Flow is claimed to preserve diversity, a quantitative comparison using standard mode-collapse or coverage metrics (beyond FID) would help substantiate this claim.

**Questions:**

1. How sensitive is pi-Flow performance to the policy horizon (number of substeps)? Could adaptive or learned horizons improve efficiency?
2. What are the computational trade-offs between DX and GMFlow in practice—does GMFlow’s expressivity justify its added complexity?

---

> ### Author Response · Authors · 2025-11-28
>
> We thank the reviewer for the positive comments on novelty and strong empirical results.
>
> **Formal analysis of error bounds or convergence**
>
> Thanks for pushing for theoretical rigor. Pi-Flow is a direct instantiation of standard on-policy imitation learning in the DAgger style, so it inherits the error bounds and convergence of DAgger. In the revised paper (L251-256), we added this paragraph:
>  > As discussed by Ross et al. (2011), on-policy imitation learning guarantees that the performance of the learned policy is bounded by the teacher's performance plus an error term that scales as $O(n\varepsilon)$, where $n$ is the number of substeps and $\varepsilon$ is the average imitation error (velocity error $\times$ substep size), which is strictly better than the $O(n^2\varepsilon)$ compounding-error behavior of off-policy behavior cloning. Moreover, the sequence of on-policy iterates converges in performance to the best policy in the function class, under the student's capacity constraint.
>
> **Do gains arise from the pi-Flow framework itself or the added modeling flexibility of GMFlow?**
>
> The gains mostly arise from the pi-Flow framework, though better policies may have an edge on smaller models that are limited in network capacity.
>
> This can be inferred from the ablation studies in Table 1 and Table 5. Both DX and GMFlow policies work reasonably well, though the differences in performance are larger in smaller models (ImageNet FID: 3.07 (GM) vs 4.44 (DX)) than in larger models (FLUX FID: 14.3 (GM) vs 14.9 (DX)). On the largest Qwen-Image model, the two policies are effectively on par (Qwen FID: 12.8 (GM) vs 12.7 (DX)). This demonstrates that the pi-Flow framework is generally effective, especially for larger models.
>
> **Tuning cost of additional policy hyperparameters**
>
> Pi-Flow, especially the GMFlow policy, requires minimal tuning efforts. This is evident in the ablation studies in Table 1, which shows that GMFlow is not sensitive to the number of Gaussian components. Therefore, for the text-to-image models, all GMFlow policy hyperparameters (8 Gaussians, 128 total integration substeps, LxC factorization) follow the exact same setting in the original GMFlow paper without further tuning.
>
> **Quantitative metric for diversity**
>
>  We’d like to clarify that FID is NOT used as the diversity metric. In Table 4, we measure the diversity using the bespoke *Diversity* metric proposed in OneIG-Bench (NeurIPS 2025), which is calculated from the similarity scores of 4 images generated from the same prompt. The quantitative results show that pi-Flow has significantly better diversity than other 4-NFE methods, aligning with the qualitative results.
>
> **Sensitivity of pi-Flow performance to the policy horizon. Could adaptive or learned horizons improve efficiency?**
>
> The performance of pi-Flow is not sensitive to the number of substeps. We tried doubling the substeps (128 -> 256) and the metrics are nearly the same. Moreover, the number of substeps have minimal impact on the efficiency. In the updated paper (Table 6 in page 10), we show that policy integration (128 total substeps) only introduces 3% overhead compared to the network evaluation cost (4 NFEs). System-level optimizations (e.g., triton/CUDA kernels instead of python for-loop integration) can further reduce the substep integration cost. Therefore, we believe there is no immediate need to apply adaptive horizons or learnt horizons.
>
> **Computational trade-offs between DX and GMFlow**
>
> In the updated paper (Table 6 in page 10), we show that DX and GMFlow have almost the same inference time, and both are minimal and negligible compared to network evaluation, so a trade-off does not exist. The seemingly complex math of GMFlow is currently handled by torch.jit compilation that fuses multiple math operations into one, making it very efficient (< 0.5 ms per policy substep) on modern hardware.
>
> **We hope that our replies have addressed the questions from the reviewer, and we kindly ask the reviewer to consider raising the score.**

---

### Official Review · Reviewer_3qdq · 2025-11-01

**Soundness:** 3
**Presentation:** 3
**Contribution:** 3
**Rating:** 8
**Confidence:** 3

**Summary:**

This paper proposes pi-Flow, a policy-based paradigm for distilling multi-step flow-matching teachers into few-NFE students. Instead of predicting a single shortcut velocity, the student network outputs a network-free policy that can be rolled out with many cheap sub-steps to integrate the probability-flow ODE. Training uses on-policy imitation distillation (pi-ID), a velocity matching loss applied along the student’s own trajectory. Two policy families are introduced—DX and GMFlow. Experiments demonstrate state-of-the-art 1-NFE FID on ImageNet-256 with DiT-XL/2 and high-quality 4-NFE text-to-image generation from 12B/20B teachers (FLUX.1, Qwen-Image), with notably better diversity than prior few-step methods.

**Strengths:**

1. By predicting a policy rather than a shortcut, pi-Flow retains the dense ODE trajectory of the teacher while requiring only a few network evaluations. This insight is both conceptually clean and practically powerful.
2. The velocity matching on the student’s own rollout avoids the complex progressive/consistency schedules of prior work and directly inherits teacher quality/diversity.
3. Strong empirical results across scales. 1-NFE FID 2.90 on ImageNet-256 (DiT-XL/2) beats all prior DiT-based few-step models. 4-NFE distillation of 12B/20B T2I models preserves fine details and text rendering while achieving substantially better diversity than SOTA few-step baselines.

**Weaknesses:**

1. Table 1 shows K=8 vs. K=32 gives similar FID, but no ablation on $L\times C$ factorization, mixture renormalization, or temperature scaling in inference.

**Questions:**

1. How does pi-Flow compare to MeanFlow in terms of training stability and wall-clock time? Does avoiding JVP give measurable speedups during training?

---

> ### Author Response · Authors · 2025-11-28
>
> We thank the reviewer for the positive comments on novelty and strong empirical results.
>
> **Ablation on LxC factorization**
>
>  We added a qualitative ablation study that explores alternative factorizations in the updated appendix (L1105-1113 and Fig. 11 in page 21). The results show that pixel-wise factorization achieves the best results with neutral colors and rich textures, aligning with the design choice in the GMFlow paper.
>
> **Ablation on mixture renormalization**
>
> It’s not fully clear to us what the reviewer means by mixture renormalization. If it refers to the normalization constant (denominator) of GMFlow mixture weights, it’s basically standard softmax operation (the mixture weights are equivalent categorical probabilities). Such an operation is standard practice. We are happy to make further replies if the reviewer can make further clarifications.
>
>  **Ablation on temperature scaling in Table 1**
>
> We did not use temperature scaling for the ImageNet models. Our ImageNet models are minimal implementations with only GM dropout and no other techniques.
>
> Temperature scaling is only adopted in text-to-image models, and we include an ablation study in the appendix (Table 7, in page 17 of the updated paper).  The results indicate that reduced temperature yields better FID (14.3 vs 14.9) on FLUX.
>
> **How does pi-Flow compare to MeanFlow in terms of training stability and wall-clock time?**
>
>  Great question!
>
>  Regarding stability, we did not encounter any stability issues with the GMFlow policy throughout the experiments, and its training dynamics is highly similar to flow matching (FM) pretraining. In contrast, MeanFlow can be unstable sometimes:
>  - Community implementation of MeanFlow reports instability when finetuning an existing FM model, as described in [1]:
>    > Direct fine-tuning using MeanFlow with classifier-free guidance (CFG) exhibits training instability. To address this issue, we adopt a staged training strategy: initially fine-tuning with MeanFlow without CFG for 20 epochs, followed by continued fine-tuning with CFG-enabled MeanFlow.
>  - We also tried distilling the CFG-enabled FM teacher using MeanFlow but failed to make it converge: the results oscillate between blurry images and noisy patterns.
>
>  Regarding training time, under PyTorch implementations of ImageNet DiT distillation, pi-Flow is about 60% faster than MeanFlow per iteration. The total wall time is also dependent on the training schedule. Concurrent work FACM (JVP-based) reports 100 epochs of distillation for the 2-NFE model, whereas pi-Flow uses around 77 epochs for the 2-NFE model. Therefore, the total wall time of pi-Flow is about half of the JVP-based FACM (not counting its additional auxiliary loss).
>
>  [1] https://github.com/zhuyu-cs/MeanFlow

---

### Author Response · Authors · 2025-11-28
**Summary of paper revision**

We sincerely thank all the reviewers for the positive comments on the novelty of the method and the strong empirical results on large scale text-to-image generation. To address the raised concerns, we have updated the manuscript with the following changes.

**pi-Flow efficiency and inference time report**

In the updated manuscript (L481-485 and Table 6), we show that pi-Flow’s total overhead is only 3% compared to other few-step models, and DX and GMFlow policies do not show measurable differences in inference time. Therefore, the inference efficiency of pi-Flow is on-par with other few-step models such as Qwen-Image Lightning.

This also indicates that pi-Flow is highly efficient in training (in terms of training time per student update iteration) when compared to other distillation frameworks: pi-Flow training effectively needs only one forward and one backward pass of the student. In contrast, JVP-based consistency distillation (e.g. MeanFlow) requires an additional JVP forward pass of the student (in our response to Reviewer 3qdq, we showed that the wall time of pi-Flow is roughly half of the JVP-based concurrent work FACM), and VSD (DMD) methods require additional fake score forward and backward passes.

**Discussion on error bounds and convergence**

In the updated manuscript (L251-256), we added a brief discussion on the theoretical error bounds and convergence of pi-ID:
 > As discussed by Ross et al. (2011), on-policy imitation learning guarantees that the performance of the learned policy is bounded by the teacher's performance plus an error term that scales as $O(n\varepsilon)$, where $n$ is the number of substeps and $\varepsilon$ is the average imitation error (velocity error $\times$ substep size), which is strictly better than the $O(n^2\varepsilon)$ compounding-error behavior of off-policy behavior cloning. Moreover, the sequence of on-policy iterates converges in performance to the best policy in the function class, under the student's capacity constraint.

**Ablation on LxC factorization**

In response to Reviewer 3qdq, we added a qualitative ablation study that explores alternative factorizations in the updated appendix (L1105-1113 and Fig. 11 in page 21). The results show that pixel-wise factorization achieves the best results with neutral colors and rich textures, aligning with the design choice in the GMFlow paper.

**Intuitive explanation of GMFlow Policy**

Although GMFlow has been extensively described in the prior work[1], we followed the suggestion of Reviewer 1tXL and added a brief intuitive explanation in the paper (L197-199):
> Intuitively, the student network $G_\phi$ maps the initial state $x_{t_\text{src}}$ to multiple denoising modes that parameterize the GMFlow policy. The policy then enables a closed-form velocity expression at future state $(x_t, t)$ for any $0 < t < t_\text{src}$.

**Additional changes**

In addition, we also self-reviewed and further polished the writing of the paper. Some ImageNet FIDs are improved slightly, since we found a minor bug in our evaluation code.

---

### Meta-Review · Area_Chair_jArP · 2025-12-23

**Summary:**

This paper introduces pi-Flow, a novel few-step distillation paradigm that replaces standard shortcut predictions with a network-free policy, achieving high quality, efficiency as well as diversity. In the initial review, reviewers rated this paper with scores of 8, 4, 4, and 8. Primary concerns involved the error bounds and convergence, efficiency overhead of policy, the comparison with different models and backbones, and the clarification to GMFlow.

During rebuttal, authors provides response to each question and most concerns are addressed. Although Reviewer KQWK and EmE8  do not have the chance to reply to the authors, AC confirms that their concerns in the initail review are mostly addressed. Since the experiment is extensive, and the results are impressive,  AC recommends this paper to Accept (Poster).

**Reviewer Concerns:**

The initial concerns involved the error bounds and convergence, efficiency overhead of policy, the comparison with different models and backbones, and the clarification to GMFlow.

During the rebuttal, the authors provided a formal theoretical analysis showing that their on-policy  imitation learning method achieves a strictly better error bound of $O(n\epsilon)$ compared to off-policy methods. They also included an inference time report, demonstrating that policy integration adds only a 3% computational overhead. The authors emphasize the comparision with DiT models (Flux and Qwen-Image), UNet models (SDXL), and non-flow matching models (MeanFlow). They also clarified the difference from GMFlow.  Consequently, AC confirms that the reviewers' concerns in the initail review are mostly addressed..

**Reviewer Scores:**

**Reviewer 3qdq**: keep 8. Reviewer 3qdq does not participates in the discussion, but his concerns and questions are all responded by the authors. AC thinks his concerns are mostly addressed.

**Reviewer KQWK**: raise to 6. Reviewer KQWK does not participates in the discussion, but his concerns and questions are all responded by the authors. AC thinks his concerns are mostly addressed.

**Reviewer EmE8**: raise to 6. Reviewer EmE8 does not participates in the discussion, but his concerns and questions are all responded by the authors. AC thinks his concerns are mostly addressed.

**Reviewer 1tXL**: keep 8. Reviewer 1tXL does not participates in the discussion, but his concerns and questions are all responded by the authors. AC thinks his concerns are mostly addressed.

---

### Decision · Program_Chairs · 2026-01-26

Accept (Poster)